# POSH: Using Scene Graphs to Guide LLMs-as-a-Judge for Detailed Image Descriptions

**Amith Ananthram**◇, **Elias Stengel-Eskin**⋆, **Lorena A. Bradford**♣,
**Julia Demarest**♣, **Adam Purvis**♣, **Keith Krut**♣, **Robert Stein**♣,
**Rina Elster Pantalony**‡, **Mohit Bansal**†, **Kathleen McKeown**◇

◇Columbia University    ⋆The University of Texas at Austin
♣The National Gallery of Art    ‡UCLA    †UNC Chapel Hill
amith@cs.columbia.edu

## ABSTRACT

While vision-language models (VLMs) have advanced into detailed image description, evaluation remains a challenge. Standard metrics (e.g. CIDEr, SPICE) were designed for short texts and tuned to recognize errors that are now uncommon, such as object misidentification. In contrast, long texts require sensitivity to attribute and relation attachments and scores that localize errors to particular text spans. In this work, we introduce POSH, a metric for detailed image description that uses scene graphs as *structured rubrics* to guide LLMs-as-a-Judge, producing aggregate scores grounded in fine-grained errors (e.g. mistakes in compositional understanding). POSH is replicable, interpretable and a better proxy for human raters than existing metrics (including GPT4o-as-a-Judge). To validate POSH, we introduce a new dataset, DOCENT. This novel benchmark contains artwork, paired with expert-written references, and model-generated descriptions, augmented with *granular* and *coarse* judgments of their quality from art history students. Thus, DOCENT enables evaluating both detailed image description metrics and detailed image description itself in a challenging new domain. We show that POSH achieves stronger correlations (+0.05 Spearman $\rho$) with the human judgments in DOCENT than the best open-weight alternatives, is robust to image type (using CapArena, an existing dataset of web imagery) and is a capable reward function, outperforming standard supervised fine-tuning. Then, using POSH, we characterize the performance of open and closed models in describing the paintings, sketches and statues in DOCENT and find that foundation models struggle to achieve full, error-free coverage of images with rich scene dynamics, establishing a demanding new task to gauge VLM progress. Through both POSH and DOCENT, we hope to enable advances in important areas such as assistive text generation. We make our metric and our benchmark available at https://github.com/amith-ananthram/posh.

## 1 INTRODUCTION

A picture is worth a thousand words – can vision-language models (VLMs) capture all of them? VLMs have saturated traditional image understanding benchmarks from short captioning to question answering (Li et al., 2025). New, more challenging tasks are needed to measure VLM progress. Detailed image description is of particular interest as it requires *comprehensive* understanding – e.g., in Fig. 1, a VLM must correctly specify *who* is pouring the water. This deep perception is a better proxy for the demands of the real world, where diverse user queries may not be reflected in VQA benchmarks (Chen et al., 2024). Moreover, it enables meaningful applications such as image assistive ("alt") text generation that could greatly expand accessibility online (Mack et al., 2021).

However, making progress on detailed description requires cheap, reliable methods for scoring models. Human evaluation is costly, involving the painstaking comparison of long texts. Even so, there is often no substitute as most metrics were designed for short texts and older models (Berger et al., 2024). Moreover, while metrics that produce a single coarse score of overall quality allow for the

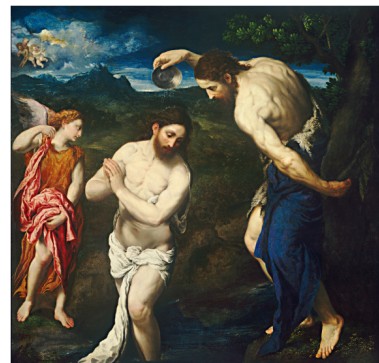

**Description A** (*better*)

...In the center stands a figure draped in white cloth around the waist, shown in a humble, **bowed posture**. To the right, a muscular figure wearing blue robes holds a vessel and is pouring water. On the left, a figure in vibrant orange....

**Description B** (*worse*)

...with three figures, **all** nude. The central figure is a man who appears to be pouring water from a vessel **into a basin**... To his right, there is another man who is **seated on a rock**... On the left side, a woman...

| | |
|---|---|
| BLEU | B |
| METEOR | B |
| CIDEr | B |
| Bert-S | B |
| SPICE | B |
| CAPTURE | B |
| LLM-as-a-Judge | B |
| PoSh 🪆 | A |

Figure 1: Failures in attribute/relation attachment are common in detailed image description, especially in dynamic scenes. Here, the *man pouring water* is not *central*. POSH catches such errors.

ranking of models, they offer little insight into the granular issues driving performance. Granular issues include *mistakes* in each generation, like the positions of the people in Fig. 1, and *omissions* in each reference, like the details of the bird's beak in Fig. 3. Automatically localizing such errors is critical as long generations with similar coarse scores may differ in multiple dimensions of interest (e.g., facial features, body orientations, etc.). Otherwise, prompt and/or model iteration necessitates expensive manual inspection to understand which description aspects need improvement.

In this work, we propose POSH[1], a metric for evaluating detailed descriptions that addresses these challenges. POSH extracts scene graphs from a generated description and its reference to use as *structured rubrics* for an LLM to granularly identify mistakes and omissions (see Fig. 2), pinpointing the textual spans containing errors like attribute/relation mis-attachment. Then, it aggregates these localized errors into coarse scores for mistakes, omissions and overall quality. Thus, POSH weds the strengths of structured methods like scene graphs (Anderson et al., 2016), which reduce descriptions to their consequential visual components, with the strengths of LLMs/VLMs-as-a-Judge (Zheng et al., 2023), which flexibly compare these visual components against diverse surface realizations.

As POSH's coarse scores are grounded in its granular scores, it is interpretable, providing clear insights into the errors driving model performance. Moreover, because POSH is entirely open-weight, it is inexpensive to use and perfectly replicable, an important pre-requisite for both adoption by researchers and deployment by practitioners that is not afforded by closed models.

Efforts to introduce metrics for longer generations have been constrained by a lack of human judgments, especially at a granular scale and for diverse imagery (see Table 1). To address this, we introduce DOCENT, a novel benchmark whose focus is visual art. DOCENT contains paintings, sketches and sculptures with expert-written assistive text that exhaustively describes features like clothing, physical orientation, relative positioning and gaze, drawn from the U.S. National Gallery of Art (see Figs. 2 and 3). It includes generations from current VLMs with judgments from art history students of their mistakes, omissions and overall quality at two resolutions: granular and coarse. Thus, DOCENT enables evaluating description[2] metrics and descriptions themselves.

We validate POSH against the human judgments in DOCENT. We show that POSH recovers human description rankings more often (+3 percentage points) and achieves stronger correlations with human-derived scores (+0.05 Spearman $\rho$) than existing overlap and open-weight alternatives (e.g. SPICE, CAPTURE, LLaVa-Critic), even surpassing GPT4o-as-a-Judge. Moreover, using judgments in CapArena (Cheng et al., 2025), we show this strength is robust to image type. Then, given its calibration, we experiment with using POSH as a reward function for describing the images in DOCENT and find that this yields meaningfully better descriptions than supervised fine-tuning (SFT).

---

[1] POSH (PrOofing Scene grapHs) can judge if your detailed descriptions are what you (really really) want.

[2] AI research often uses *caption* and *alt-text* interchangeably. However, according to Web Content Accessibility Guidelines, *captions* are related to an image while *alt-text* conveys the information in an image. As our focus is evaluating generations that could serve as *alt-text*, we use the term *description*.

Finally, using POSH, we characterize the performance of open and closed models in describing the artwork in DOCENT, establishing a difficult new task. In so doing, we extend detailed description to a technically challenging and socially impactful domain: assistive text generation for artwork, whose visual complexity and diversity stress VLMs (Bengamra et al., 2024) (see Fig. 1).

In summary, our contributions are:

1. We propose POSH, a new metric for detailed description evaluation. POSH is interpretable, producing *coarse* scores grounded in *granular* scores that are localized to text spans.

2. We present DOCENT, a new detailed description benchmark with 1,750 expert-written art descriptions and 900 *granular* & *coarse* judgments of generations from informed raters.

3. We show POSH correlates more with DOCENT's judgments than existing metrics and GPT4o while being replicable. On CapArena, we confirm POSH is robust to image type.

4. We demonstrate that using POSH as a reward function outperforms SFT on DOCENT.

5. Using POSH and DOCENT, we evaluate both open and closed models on detailed description of artwork, establishing a socially impactful new task to gauge VLM progress.

## 2 RELATED WORK

Image description is under-specified – the correct way to describe an image is often task-specific. This is especially true for assistive text which has context-dependent requirements (Kreiss et al., 2022). Moreover, in such sensitive applications, correlated failures between reference-free metrics and VLMs relying on similar components could prove dangerous to end users (Deutsch et al., 2022). Thus, our focus is reference-based evaluation. Traditional metrics were not designed to evaluate long text and can involve truncation due to limited context length (e.g. CLIPScore) (Papineni et al., 2002; Lin, 2004; Banerjee & Lavie, 2005; Vedantam et al., 2015; See et al., 2017; Hessel et al., 2021; Sarto et al., 2023). Recent work has explored LLMs/VLMs-as-Judges though this requires potentially expensive API calls and offers limited replicability (Chan et al., 2023; Cheng et al., 2025). Even when replicable, they do not provide interpretable, grounded granular scores (Xiong et al., 2025).

While prior metrics like SPICE and CAPTURE leverage scene graphs, they forgo their rich structure by ignoring object attachment (Anderson et al., 2016; Dong et al., 2024). This favors generations with misattributed details (as in Fig. 1). In summarization, Scialom et al. (2021) use question generation and answering (QA) to compare a summary and its source. In text-to-image generation, Cho et al. (2024) use GPT4 to extract and verify a scene graph from a visual prompt. POSH extends these approaches to detailed description evaluation that is replicable and interpretable. With small models, it extracts scene graphs to use as structured rubrics for guiding an open-weight LLM-as-a-Judge.

Table 1: Detailed image description benchmarks with summaries of their images, reference descriptions (where detail is average # of entities + attributes + relations) and judgments (where source is the type of annotator used and time is the average time per judgment). Most benchmarks release no human judgments. In contrast, DOCENT contains both granular and coarse judgments of long descriptions of visually complex artwork elicited from annotators knowledgeable in art.

| Name | Images | Reference Descriptions | | | Judgments | | | |
|---|---|---|---|---|---|---|---|---|
| | Source | Source | Words | Detail | Source | Type | Time (min) | # |
| DCI | web | crowd | 133 | 71 | *no judgments* | | | |
| DOCCI | web | crowd | 122 | 66 | | | | |
| CompreCap | web | crowd | - | - | | | | |
| DeCapBench | uses ImageInWords | | | | | | | |
| ImageInWords | web | crowd+ | 193 | 113 | *no judgments with references*[3] | | | |
| DetailCaps | web | model | 154 | 95 | model | coarse | - | 14.4K |
| CapArena | uses DOCCI | | | | skilled | coarse | 2.4 | 5.6K |
| DOCENT (ours) | art | expert | 251 | 161 | skilled | granular | 18 | 300 |
| | | | | | | coarse | 5 | 600 |

---

[3]The judgments in IIW compare 1) paired references and 2) paired generations for images with no references. As such, they cannot be used to evaluate a reference-based metric.

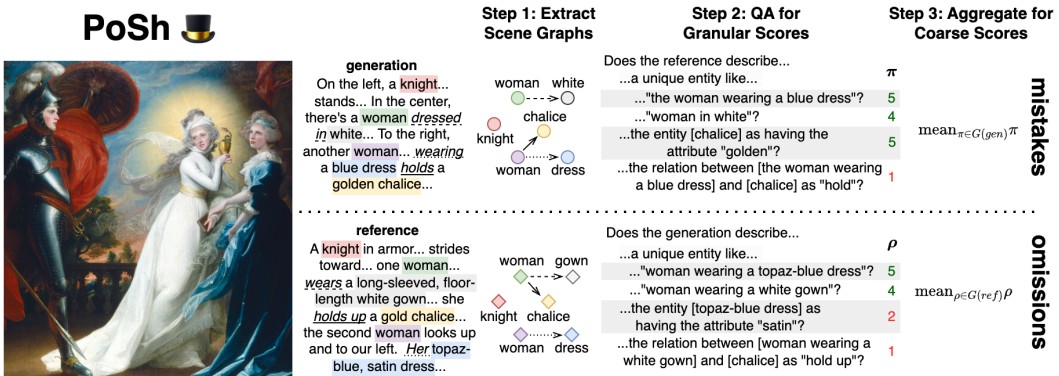

Figure 2: PoSh, a metric for detailed description evaluation, that produces granular and coarse scores. **Step 1:** Given a generated description and its reference, PoSh extracts scene graphs that reduce each text's surface diversity to its objects, attributes and relations. **Step 2:** Using each scene graph as a *structured rubric*, PoSh produces granular scores for the presence of its components in the other text through QA. **Step 3:** PoSh aggregates these granular scores for each scene graph to produce interpretable coarse scores for mistakes and omissions.

Evaluating such a metric requires human judgments of model generations. Though there are many detailed image description benchmarks (Urbanek et al., 2024; Onoe et al., 2024; Garg et al., 2024; Lu et al., 2025; Ye et al., 2025), most release no such judgments. One notable exception is CapArena which contains coarse rankings of descriptions for web imagery (Cheng et al., 2025). In contrast, our new dataset, DOCENT contains both *granular* and *coarse* judgments, enabling the evaluation of fine-grained metrics like PoSh. Moreover, it expands detailed description to artwork whose scene dynamics and expert-written references are considerably more complex (see Table 1).

## 3 PoSh: A New Metric for Detailed Image Description

PoSh is a reference-based metric for detailed image description evaluation that takes two descriptions, a generation and its reference, and then extracts scene graphs from each to use as *structured rubrics* for granular and coarse evaluation of mistakes (i.e. precision) and omissions (i.e. recall).

It does so in three steps (Fig. 2): **Step 1)** It extracts scene graphs from a generation and its reference that preserve object attachments. **Step 2)** It evaluates the presence of generation scene graph components in the reference (and reference scene graph components in the generation) through question answering with an LLM to identify granular mistakes (and omissions). **Step 3)** It produces coarse scores for mistakes and omissions grounded in these granular scores. We discuss each step below.

**Scene Graph Extraction**   As in SPICE (Anderson et al., 2016), given a description $d$, a scene graph $G(d)$ is a structured representation of $d$. Specifically, $G(d) = \langle O(d), E(d), K(d) \rangle$ where $O(d) \subseteq C$ is a set of objects, $E(d) \subseteq O(d) \times A$ is a set of attributes associated with each object and $K(d) \subseteq O(d) \times R \times O(d)$ are a set of relation edges between objects. $C$, $A$ and $R$ are open-world sets of all possible object, attribute and relation classes.

Given a generation *gen* with its reference *ref* we extract sentence-level scene graphs $G_i(gen), G_j(ref)$ for each using off-the-shelf dependency parsing and combine them via coreference resolution (Honnibal et al., 2020; Martinelli et al., 2024). This produces scene graphs with full coverage of each text where each component is localized to text spans, allowing for grounded, interpretable scoring. We provide pseudocode for this extraction in Appendix A.1.2.

**Granular Scoring**   Given a description $d$, its scene graph $G(d)$ and a different description $d'$, we apply the function $\Psi$ to every component $c \in G(d)$ to produce a score reflecting its presence in $d'$.

We implement this function via question answering. We produce templated questions for each scene graph component (object, attribute and relation) $c \in G(d)$ and prompt an open-weight LLM to quantify the degree to which $c$ is described in $d'$. This avoids forcing an alignment between the

components of $G(d)$ and $G(d')$. For example, in Fig. 2, the reference describes the figures in the image as a "trio." Question answering ensures that a generation that refers to all three individually is not penalized for failing to include such collectives.

As objects with the same class may appear many times in a scene graph (e.g., a description of multiple men), questions require the use of unique identifiers (e.g., "woman in white" in Fig. 2) to disambiguate such instances in $d'$. As the identifier used in $d'$ (if any) is not known *a priori*, we test candidate identifiers in three passes, first considering only objects not part of any other objects in $G(d)$ (e.g., "man" but not "face of the man"), then objects that are part of other objects in $G(d)$ (e.g., "face of the man") and finally attributes and relations of objects identified as present in $d'$.

When collecting unique candidate identifiers for an object $o \in O(d)$, we consider its class name (e.g. "man"), its surface form (e.g. "musician"), its attributes (e.g. "tall man"), its relations (e.g. "man on horse") and if part of a previously identified object, its "part-of" relation (e.g. "face of tall man"). We re-write these identifiers using our LLM to improve their fluency and then test each one in bulk for their presence in $d'$. We use the simplest identifier confirmed present by our LLM (if any) to evaluate $o$'s attributes and relations. We provide pseudocode for this templating in Appendix A.1.3.

We produce granular mistake scores $\pi$ for every component of $G(gen)$ and granular omission scores $\rho$ for every component of $G(ref)$:

$$\pi(c_{gen}) = \Psi(c_{gen}, ref), \forall c_{gen} \in G(gen) \qquad \rho(c_{ref}) = \Psi(c_{ref}, gen), \forall c_{ref} \in G(ref)$$

**Coarse Scoring** To maintain interpretability, we calculate coarse scores for mistakes (i.e. precision) and omissions (i.e. recall) by averaging over our granular scores directly:

$$\text{Mistakes} = \text{mean}_{c \in O(gen)}(\pi(c)) \qquad \text{Omissions} = \text{mean}_{c \in O(ref)}(\rho(c))$$

We note this is a natural place to introduce tunable weights (as in Dong et al. (2024)) to adapt POSH to particular datasets. As we aim to demonstrate robustness, we leave these terms unweighted.

## 4 DOCENT: A NEW BENCHMARK FOR DETAILED DESCRIPTION OF ART

DOCENT is a benchmark for evaluating detailed description metrics and detailed descriptions themselves. It consists of $1,750$ works of art with expert-written references from the Open Data Program at the U.S. National Gallery of Art (NGA)[4]. For 100 of these images, we produce four generations from current small and frontier VLMs and collect 300 granular (for 75 images) and 600 coarse judgments from annotators knowledgeable in art of *mistakes* and *omissions*[5]. On average, coarse judgments took 5 minutes and granular judgments took 18 minutes (six annotation days). This highlights both the cost of manual evaluation and the need for metrics that are reliable proxies.

We include summary statistics in Table 1 and example judgments in Fig. 3.

**Image / Reference Selection** While the majority of these works are paintings, they include sketches, statues and lithographs (e.g., the bird in Fig. 3), all in the public domain. These images span a diverse set of styles (e.g., Baroque, Renaissance, Impressionism, Post-Impressionism), themes (e.g., war, courtship, still life, religion) and topics (e.g., fishing, drinking, animals, boating).

The accompanying references are detailed descriptions whose purpose is accessibility – as such, they follow guidelines[6] that include tips for describing color (e.g., "color can be likened to temperature") and handling ambiguity (e.g, "describe what makes something ambiguous"). These context informed requirements highlight the need for reference based metrics (Kreiss et al., 2022).

Compared to existing detailed image description benchmarks, DOCENT contains considerably more visual complexity (see Table 1). On average, its images contain $16\%$ more objects and nearly twice as many people[7] who require description of their orientation, features, clothing, etc. Consequently, the average length and scene graph size of its reference descriptions are nearly double.

---

[4] https://www.nga.gov/open-access-images/open-data.html
[5] We forgo fluency as recommended by Kasai et al. (2022)
[6] www.nga.gov/visit/accessibility/collection-image-descriptions
[7] As measured by OneFormer Jain et al. (2023)

Figure 3: DOCENT, our newly introduced benchmark, is the first to contain both *granular* (top) and *coarse* (bottom) judgments from informed raters of detailed descriptions of artwork.

**Model Selection** We generate detailed descriptions for 100 images in DOCENT from four current VLMs that span transparency and model size (from open data/open weight to frontier models): LLaVA-1.6-7B (Liu et al., 2024), Molmo-D-7B (Deitke et al., 2024), GPT4o and Claude Sonnet 3.5. A metric that discriminates among these generations similarly to their human judgments could gauge progress in detailed image description in small and large VLMs over time. Additional details (prompts, date of API access) can be found in Appendix A.2.1.

**Annotators** Given the complexity of our images and the detail of their expert descriptions, we recruit 24 art history undergraduate majors, masters students and PhD students with domain familiarity to provide high quality judgments of generations. All annotators were sighted with full color vision and native speakers of English. They were compensated at a rate of $22/hour for their time.[8]

**Granular Judgments** Half of our annotators identify *mistakes* and *omissions* in our model generations. For each image, an annotator is shown its reference and then its four model generations in random order. First, they look at the image, read the reference and then the current generation. Next, by selecting narrow text spans, annotators first identify *mistakes* in the generation (i.e. precision errors) and then *omissions* in the reference that are not in the generation (i.e. recall errors). When identifying omissions, as in Kasai et al. (2022), we ask annotators to mentally correct narrow mistakes in the generation first to avoid double-penalizing a model for both incorrect specificity and lack of specificity. For example, a generation that describes a *woman* as a *man* is an error in precision but not in recall. We include our task instructions and interface[9] screenshots in Figs. 4 and 6.

**Coarse Judgments** The other half of our annotators provide coarse judgments of our model generations. For a given image, an annotator is shown its reference and two generations (#1 and #2) in random order and asked to rank the generations in terms of mistakes (i.e. precision), omissions (i.e. recall) and overall quality. These pairwise judgments avoid some of the inter-annotator inconsistency of Likert ratings, especially for long text (Novikova et al., 2018).

Annotators select among five choices for each dimension: #1 much better, #1 slightly better, equal, #2 slightly better and #2 much better. As with our granular judgments, we ask annotators to mentally correct narrow mistakes (i.e. precision errors) in each generation before judging omissions. To avoid favoring previously seen generations, we ensure no annotator sees the same generation more than once. We include our task instructions and screenshots of our annotation interface[9] in Figs. 5 and 7.

---

[8]This study was conducted under Columbia University IRB protocol AAAV6216.

[9]Hosted on Label Studio (https://labelstud.io, Tkachenko et al. (2020-2025))

**Agreement** For a given image, each generation / pair of generations receives at least one granular and one coarse judgment respectively. For $15\%$ of our tasks, we collect additional judgments from our annotators (2 for coarse, 1 for granular). Additionally, for 20 granular tasks and 30 coarse tasks, we collect expert judgments from a PhD in art history who authors assistive text at an art museum. We use these extra judgments to calculate agreement in two ways (among our annotators and between our annotators and our expert). We report agreement in Tables 4 and 5 of the Appendix.

For our granular judgments, as recommended by Hripcsak & Rothschild (2005) for span annotation tasks where the boundaries of negative examples (i.e. non-errors) are ill-defined, we measure agreement using the relaxed F1 (matching spans that contain 50% overlapping tokens). Under this measure, our student annotators exhibit strong agreement among themselves and with our expert.

Our coarse judgments exhibit moderate inter-annotator agreement, with Krippendorf's $\alpha = 0.509$, $0.409$ and $0.459$ for mistakes, omissions and overall quality (Landis & Koch, 1977). This level of agreement is unsurprising for coarse detailed description evaluation – judgment requires weighing the relative importance of each text's granular errors and is consequently more subjective. Nevertheless, our student annotators exhibit moderate to strong correlations with our expert, with significant Pearson $\rho$ values of $0.727$, $0.501$ and $0.492$ for mistakes, omissions and overall quality respectively.

**How well do these VLMs describe art?** When considering the performance of the four models included in DOCENT, we observe expected trends, adding to our confidence in the quality of our judgments: the smaller models make more mistakes and have more omissions than the larger models (see Tables 4 and 5). Though most models make few mistakes, they all struggle with omissions. The best model, `gpt4o` covers only $50.1\%$ of the visual information conveyed in DOCENT's references. Raising this requires continued prompt iteration, highlighting the need for an automated metric that can reliably measure both granular and coarse differences in mistakes and omissions.

## 5 EXPERIMENTS

**POSH** We extract sentence-level scene graphs using `en_core_web_trf` from Honnibal et al. (2020), a transformer trained to perform dependency parsing. To merge objects across these scene graphs while preserving attribute and relation attachments, we use `maverick-mes-ontonotes` from Martinelli et al. (2024) to perform co-reference resolution. Our QA scorer $\Psi$ is `qwen-3-14b` (Yang et al., 2025). We template evaluation questions for each scene graph component (as in Fig. 2), re-write candidate identifiers using $\Psi$ to improve fluency and then prompt $\Psi$ to answer each templated presence question by predicting a number between $1$ and $5$. We extract scores by taking the weighted average over the token logits for each number as in Liu et al. (2023). When determining object presence, we use a threshold of 2, determined through tuning on a small hand-annotated validation set. We provide further implementation details and all prompts used in Appendix A.1.

**Benchmarks** We evaluate POSH against the judgments in DOCENT and CapArena.

*DOCENT* is our new detailed description benchmark containing judgments from knowledgeable human annotators: granular mistake and omission spans for 300 individual generations and coarse scaled rankings of mistakes, omissions and overall quality of 600 paired generations. We evaluate granular metrics on this benchmark using macro F1 where we credit/penalize a model for predicting each annotated/unannotated token. Our coarse judgments are in the form $(text_1, text_2, score)$ where score indicates how much better or worse $text_1$ is than $text_2$. We evaluate each coarse metric $m$ by calculating its 1) pairwise accuracy (whether it picks the better text or a tie, using a tie threshold inferred from the gold tie rate) and 2) Spearman rank $\rho$ and Kendall's $\tau$ correlations between $m(text_1) - m(text_2)$ and $score$, a common practice in machine translation metric evaluation (Kocmi et al., 2021). More details can be found in Appendix A.3.1.

*CapArena* (Cheng et al., 2025) contains $3,361$ images and $10,348$ detailed descriptions generated from 14 current VLMs. $5,599$ pairs of these generations receive coarse judgments from human raters of the better generation (or "tie"). We include CapArena, which contains diverse images drawn from the web, to validate metric robustness. However, we note the dramatic simplicity of its images and references compared to those in DOCENT (see Table 1). $64\%$ of its images[10] contain fewer than

---

[10]As measured by OneFormer (Jain et al., 2023)

two objects and $95\%$ depict fewer than two people (compared to $27\%$ and $52\%$ in DOCENT). A metric is evaluated on CapArena at the caption-level (whether it picks the better text or a tie, using a tie threshold inferred from the gold tie rate) and at the model-level (Spearman's rank and Kendall's $\tau$ correlation between ELO rankings derived from metric predictions and gold judgments).

**Granular Baselines** Our work is the first to introduce both a metric and a benchmark for granular evaluation of detailed descriptions. As such, this limits our baselines to those able to predict *localized* mistakes and omissions (i.e., the spans where errors occur). We consider two embedding-based approaches, using `Qwen/Qwen3-Embedding-8B` from Yang et al. (2025): **4GramEmbed**, which embeds and compares 4-grams from a generation and its reference, and **SGEmbed**, which embeds and compares components from the scene graphs of a generation and its reference. As these approaches (and POSH) produce span scores, we report the maximum F1 scores for mistakes and omissions across all alerting thresholds. More details can be found in Appendix A.3.3.

**Coarse Baselines** Though POSH is a text-only reference-based metric, we select a representative set of reference-free (requiring only an image) and reference-based (requiring a gold standard) pointwise metrics (i.e. produce numerical scores) as our baselines. These include n-gram overlap metrics like BLEU (Papineni et al., 2002), ROUGE-L-Sum (See et al., 2017), METEOR (Banerjee & Lavie, 2005) and CIDER (Vedantam et al., 2015) and model-based metrics like SPICE (Anderson et al., 2016), CLIPScore (Hessel et al., 2021) and CAPTURE (Dong et al., 2024). Additionally, we consider several LLMs/VLMs-as-a-Judge[11]: FLEUR (Lee et al., 2024), Prometheus (Kim et al., 2023), LLaVA-Critic (Xiong et al., 2025), DCScore Ye et al. (2025), Qwen-3 (Yang et al., 2025) and GPT4o/GPT5 in three settings (reference-free with image, reference-based without image and reference-based with image). More details can be found in Appendix A.3.4.

**Reward Function** Finally, given the potential of a well-calibrated metric as a verifier in reinforcement learning (RL), we evaluate POSH as a reward function. We train `Qwen2.5-VL-7B` on the $1,000$ images in DOCENT's training set in two settings: 1) supervised fine-tuning (SFT), and 2) RL with DAPO (Yu et al., 2025) using POSH. We collect coarse judgments (as in Section 4) for $40$ generation pairs from graduate students in NLP. More details can be found in Appendix A.3.5.

## 6 RESULTS & DISCUSSION

Table 2: Granular metrics evaluated on DOCENT. Reported numbers are the maximum F1 when identifying mistakes and omissions across all alerting thresholds. POSH is best at predicting both mistakes (which are relatively rare) and omissions (which are relatively common). As POSH's coarse scores are aggregated from its granular scores, this demonstrates its interpretability.

|  | Random | 4GramEmbed | SGEmbed | **POSH** |
|---|---|---|---|---|
| Mistakes F1 | 0.503 | 0.483 | 0.514 | **0.580** |
| Omissions F1 | 0.499 | 0.641 | 0.658 | **0.680** |

**POSH as a Granular Metric** Table 2 presents the performance of POSH and our selected metrics on identifying the mistakes and omissions in DOCENT. Given the imbalanced nature of our data (where mistakes are infrequent and omissions are common), we report macro averages for each subtask, measuring how well each approach localizes errors within a generation and its reference respectively. First, we note that this task is difficult. The considerable room for improvement highlights the value of a benchmark like DOCENT that contains granular judgments of textual spans. Even so, **POSH achieves the highest F1 in mistake** (0.580**) and omission** (0.680**) localization**. As its coarse scores are aggregated from these granular scores, this demonstrates its interpretability.

---

[11]CLAIR/Faithscore were not included due to complications with their codebases (Chan et al., 2023; Jing et al., 2024). Due to cost (estimated at $\$1,000$), we only evaluate DCScore (Ye et al., 2025) on DOCENT.

Table 3: Selected coarse metrics evaluated on DOCENT and CapArena, identified with $\Theta$ (parameter count, in billions), 📄 (requires a reference), 🖼 (requires an image) and 🔁 (replicable). "acc" indicates accuracy at predicting the better generation (or "tie") in each judged pair. For DOCENT, $\rho$ / $\tau$ indicate the Spearman rank / Kendall's $\tau$ correlations between differences in the metric and differences in the rank of the generations in each pair. For CapArena, $\rho$ / $\tau$ indicate the Spearman rank / Kendall's $\tau$ correlations between model ELO rankings derived from metric scores and human judgments. **Bold** indicates the best replicable metric while underlining indicates the best metric overall. Gray cells indicate correlations that are *not* statistically significant at $\alpha = 0.05$. POSH beats nearly all baselines, including GPT4o, across both benchmarks in all settings (caption ranking of mistakes, omissions and overall quality & model ranking) while remaining perfectly replicable.

| | | | | | DOCENT | | | | | | | | | CapArena | | |
|---|---|---|---|---|---|---|---|---|---|---|---|---|---|---|---|---|
| | | | | | Mistakes | | | Omissions | | | Overall Quality | | | Desc | Model | |
| | $\Theta$ | 📄 | 🖼 | 🔁 | acc | $\rho$ | $\tau$ | acc | $\rho$ | $\tau$ | acc | $\rho$ | $\tau$ | acc | $\rho$ | $\tau$ |
| length | | | | ✓ | 30.5 | −0.270 | −0.206 | 37.8 | −0.002 | −0.001 | 38.0 | −0.160 | −0.121 | 58.7 | 0.710 | 0.582 |
| SPICE | | ✓ | | ✓ | 41.3 | 0.308 | 0.234 | 55.0 | 0.464 | 0.360 | 58.5 | 0.458 | 0.349 | 41.7 | 0.275 | 0.231 |
| CAPTURE | | ✓ | | ✓ | 43.3 | 0.259 | 0.194 | 53.8 | 0.447 | 0.340 | 56.0 | 0.453 | 0.347 | 52.5 | 0.613 | 0.538 |
| Qwen3 | 32 | ✓ | | ✓ | 57.7 | 0.282 | 0.235 | 53.5 | 0.286 | 0.253 | 61.2 | 0.289 | 0.257 | 56.2 | 0.899 | 0.714 |
| LLaVa Critic | 72 | ✓ | ✓ | ✓ | 62.8 | 0.412 | 0.351 | 53.7 | 0.509 | 0.430 | 66.8 | 0.546 | 0.461 | 64.0 | 0.987 | 0.934 |
| DCScore | | ✓ | ✓ | | 62.8 | 0.541 | 0.422 | 54.0 | 0.395 | 0.298 | 62.8 | 0.471 | 0.362 | - | - | - |
| GPT4o | | ✓ | ✓ | | 58.5 | 0.484 | 0.396 | 56.0 | 0.380 | 0.303 | 67.3 | 0.510 | 0.402 | 55.4 | 0.890 | 0.802 |
| GPT5 | | ✓ | | | 62.5 | 0.511 | 0.423 | 53.2 | 0.421 | 0.332 | 68.0 | 0.540 | 0.440 | 59.1 | 0.956 | 0.846 |
| POSH | 14 | ✓ | | ✓ | 60.7 | **0.519** | **0.405** | 62.7 | **0.581** | **0.451** | 70.7 | **0.599** | **0.466** | 59.2 | 0.931 | 0.796 |

**POSH as a Coarse Metric** Table 3 presents the performance of POSH and the best baselines on predicting the coarse judgments in DOCENT and CapArena (full results in Appendix A.4.1).

On DOCENT, across all three dimensions, **POSH outperforms every existing replicable metric** (i.e., metrics not reliant on an API), yielding a $0.11$ increase in Spearman $\rho$ for mistakes ($25\% \uparrow$), a $0.07$ increase for omissions ($14\% \uparrow$) and a $0.05$ increase for overall quality ($9\% \uparrow$) over the next best. It even outperforms GPT4o (in all settings) and text-only GPT5 (on omissions and overall quality). Among all metrics, DCScore (Ye et al., 2025) proves best at predicting mistakes. However, its reliance on GPT4o to extract and verify factoids fails to achieve full coverage of reference detail, underperforming in predicting omissions and overall quality. Despite employing a smaller LLM, POSH's use of dependency parsing and coreference resolution to extract scene graphs avoids this.

On CapArena, POSH achieves higher caption-level accuracies and model-ranking correlations than nearly every existing open-weight metric and GPT4o. The sole exception is LLaVa Critic, a much larger VLM-as-a-Judge (Xiong et al., 2025). This is driven in part by the simplicity of CapArena (see Table 1). On the subset of CapArena depicting three or more people (167 judgments), each of whom requires careful description, **POSH outperforms LLaVa Critic with model ranking correlations of $\rho = 0.727, \tau = 0.581$ compared to $\rho = 0.686, \tau = 0.550$**. Thus, POSH is robust to image type, excelling in visually complex cases that are of particular interest in detailed image description.

**POSH as a Reward Function** In Table 7 of the Appendix, we report annotator agreement and aggregate preferences between SFT and DAPO with POSH. In each dimension of interest, a POSH-tuned generation earns a score between $-2$ and $2$ based on how much worse or better it is than its SFT counterpart. While POSH-tuned generations had more mistakes (an average score of $-0.243$), these were incurred in service of **much fewer missing details ($+0.432$), resulting in higher overall quality ($+0.135$)**. This speaks to the strength of POSH when optimized directly. Given recent progress in generating synthetic detailed descriptions (Li et al., 2024), POSH-tuning could be freely scaled in post-training. Moreover, as POSH produces localized granular scores, it supports token-level guidance (Yang et al., 2023), an exciting direction to explore in future work.

**POSH Subcomponent Evaluation** POSH relies on two subcomponents, scene graph extraction and scene graph element verification through question answering. As downstream errors may propagate from these components, we validate each through comparison against hand annotated examples from DOCENT. We find that POSH's scene graphs are high quality, with average element F1 of $0.892$. Similarly, in verifying these scene graph elements, POSH exhibits strong alignment with human raters, with an F1 of $0.852$. We provide additional details and analysis in Appendix A.4.2.

**POSH Runtime**   A core enabler of POSH's performance and interpretability is its thorough granular evaluation. It achieves this efficiently through inference optimizations like continuous batching and prefix caching (Kwon et al., 2023). POSH scores the 400 examples in DOCENT in 15 minutes, or one every 2 seconds, on a single H100 GPU. In contrast, DCScore (Ye et al., 2025) takes upwards of 2 hours due to its heavy use of GPT4 (one description every 25 seconds). As manual evaluation takes 18 minutes per description (see Table 1), POSH effectively balances quality and cost.

## 7   DOCENT LEADERBOARD

Finally, in Fig. 9, we plot the POSH scores of VLMs in describing the art in DOCENT. While closed models like Gemini 2.5 Pro lead, open models remain competitive. Improvements will require continued iteration, informed in part by insights gained from analyzing POSH's granular scores.

## 8   CONCLUSION

We present POSH, a novel metric for detailed image description that extracts scene graphs to use as structured rubrics for guiding LLMs-as-a-Judge, providing interpretable, replicable scores. To validate POSH, we introduce DOCENT, a new benchmark with expert-written descriptions of visually complex artwork along with granular and coarse judgments of generations from knowledgeable raters. We show that POSH correlates better than other metrics with these judgments, is robust to image type and is a capable reward function. Through POSH and DOCENT, we introduce a leaderboard for a new challenging task, detailed image description of artwork. It is our hope that this work will drive progress in meaningful areas such as assistive text generation for artwork and beyond.

## 9   LIMITATIONS

Recent efforts have explored using structural priors to guide generation (e.g. Wang et al. (2025)). As these methods extract and describe structure from images, and as POSH extracts and validates structure from text, we do not expect POSH to be biased towards them in an evaluation setting. Nevertheless, as these models become publicly available, this requires experimental validation.

## 10   ETHICS STATEMENT

The judgments in DOCENT were collected under IRB protocol AAAV6216 with all annotator data anonymized and participants receiving fair compensation (at $22/hour) for their time and expertise.

All of the $1,750$ artwork images in DOCENT are in the public domain, and the expert-written reference descriptions were published by the U.S. National Gallery of Art under their Open Data Program[12] specifically for research purposes, ensuring appropriate use and attribution.

While this work aims to benefit accessibility applications for blind and low-vision users, we acknowledge that direct community involvement in the development process would strengthen future iterations. However, we note that the expert reference descriptions were written according to the National Gallery of Art's accessibility guidelines[13] which lay out best practices for assistive text.

Finally, as with other computer vision systems, this work could theoretically be applied to surveillance contexts, but our focus on detailed description does not introduce novel privacy risks beyond those inherent to existing image analysis technologies. The primary intended application—improving accessibility—aligns with beneficial societal outcomes.

---

[12]https://github.com/NationalGalleryOfArt/opendata
[13]https://www.nga.gov/visit/accessibility/collection-image-descriptions

## 11 REPRODUCIBILITY STATEMENT

A core motivation behind POSH is improving replicability in detailed image description evaluation through the introduction of a performant open-weight metric. In that spirit, we ensure full reproducibilty of our findings by:

1. including comprehensive technical details in the Appendix

2. publishing the code for both our metric and our metric evaluations at `https://github.com/amith-ananthram/posh`; this implementation supports batch invariance, ensuring perfect reproducibility of our results on an H100 GPU with CUDA 12.8

3. publishing our benchmark at `https://github.com/amith-ananthram/posh/tree/main/docent`

4. making our models and our benchmark available to the broader research community on HuggingFace

### ACKNOWLEDGMENTS

This research is being developed in part with funding from the National Science Foundation under Cooperative Agreement PHY-2229929 (the NSF AI Institute for Artificial and Natural Intelligence) and DRL-2112635 (the NSF AI Engage Institute), the Columbia Center for Artificial Intelligence and Technology (CAIT) and ONR Grant N00014-23-1-2356. Data and data science collaboration were provided by the National Gallery of Art in Washington, DC. We gratefully acknowledge use of the research computing resources of the Empire AI Consortium, Inc, with support from Empire State Development of the State of New York, the Simons Foundation, and the Secunda Family Foundation. The views, opinions and/or findings expressed are those of the authors and should not be interpreted as representing the official views or policies of the State of New York, the National Gallery of Art, the National Science Foundation or the U.S. Government.

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

# A  APPENDIX

## A.1  POSH

### A.1.1  COMPARISON TO DAVIDSONIAN SCENE GRAPH

There are several differences between the Davidsonian Scene Graph (DSG) metric (Cho et al., 2024) and POSH. While the emphasis of DSG is on evaluating text-to-image models, POSH was designed specifically for detailed image descriptions and is tailored to the unique challenges they pose.

First, the emphasis of DSG is on evaluating text-to-image models: it compares an image to scene graph elements extracted from a prompt text through visual question answering. As such, it cannot serve as a reference-based metric for evaluating detailed image descriptions. Allowing the use of references is important. Downstream tasks have context-specific requirements that can only be specified with references. This is especially true in accessibility (Deutsch et al., 2022).

Additionally, DSG was designed for image generation prompts at most three sentences long (in contrast, POSH is able to compare generations and references that are 10 - 20 sentences long); DSG prompts GPT-3.5 for atomic propositions that are not localized to text spans (in contrast, POSH grounds its coarse scores in localized granular scores with open models, allowing error visualization, better interpretability and replicability), DSG's atomic propositions do not have special handling for entity collisions (in contrast, POSH tests discriminating identifiers for colliding entities to allow unique validation of their presence), and finally, DSG applied to detailed image descriptions measures only precision, validating the presence of generation elements in its source image (in contrast, POSH measures both precision and recall, penalizing generations for omitting important details).

These differences in design and purpose become clear when evaluating DSG against the judgments in POSH, where its accuracies are near chance and its correlations are near zero (see Table 6).

### A.1.2  SCENE GRAPH EXTRACTION

While we provide the complete implementation for our scene graph extraction in our codebase, we include simplified pseudocode below:

```
def GetGraph(text):
    doc = ParseTextWithSpacy(text)
    components = ExtractComponents(doc)
    corefs = GetCorefWithMaverick(doc)

    entities, relations = [], []
    for each component:
      if IsNoun(component):
        if HasEarlierMention(component):
          UpdateExistingEntity(
            entities, component
          )
        else:
          CreateNewEntity(
            entities, component
          )

    for each component:
      if IsAdjective(component):
        UpdateAtributes(
          entities, component
        )
```

```
      elif IsVerb(component):
        UpdateVerbRelations(
          relations, component
        )
      elif IsPrep(component):
        UpdatePrepRelations(
          relations, component
        )

    return (entities, relations)
```

### A.1.3 GRANULAR QA TEMPLATING

While we provide the complete implementation for our question templating in our codebase, we include simplified pseudocode below:

```
def TemplateEntityQuestions(
  text, entities
):
  colls = GetCollisions(
    entities
  )

  questions = []
  for e in entities:
    identifiers = []
    if IsEmpty(colls):
      identifiers.add(e.text)

    for each attr in e:
      if IsUnique(attr, colls):
        identifiers.add(
          attr + e.text
        )

    if len(identifiers) > 0:
      AddToQuestions(identifiers)
      continue

    for each rel in e:
      if IsUnique(rel, colls)
        identifiers.add(
          rel.head +
          rel.text +
          rel.tail
        )

    AddToQuestions(identifiers)

  ReWriteIdentifiers(questions)

def TemplateAttrRelQuestions(
  text, entities
):
  questions = []
  for e in entities:
    for attr in e:
      AddToQuestions(
        attr, e.identifier
```

```
    )
  for rel in e:
    AddToQuestions(
      rel, e.identifier
    )
```

### A.1.4 PROMPTS

---

**Entity Identifier Rewrite Prompt (for attributes)**

Rewrite ''{entity_identifier}'' into a grammatically correct noun phrase, keeping all details. For example, ''dog small'' should be rewritten as ''the small dog''. Output ONLY the phrase.

---

**Entity Identifier Rewrite Prompt (for relations)**

Rewrite ''{entity_identifier}'' into a grammatically correct noun phrase, keeping all details. ''cat jumps on window'' should be rewritten as ''the cat jumping on the window''. Output ONLY the phrase.

---

**Verification Prompt**

```
if {precision}
    DESCRIPTION1: {target_text}

    DESCRIPTION2: {source_text}
{else}
    DESCRIPTION: {target_text}

{if entity}
    Q: Is an entity matching ''{entity_identifier}''
    (from DESCRIPTION2) mentioned in (the) DESCRIPTION(1)?
{elif attribute}
    Q: Is ''{entity_identifier}'' (from DESCRIPTION2)
    described as ''{attribute}'' in (the) DESCRIPTION(1)?
{else}
    Q: Is the relation between ''{entity1_identifier}''
    and ''{entity2_identifier}'' (in DESCRIPTION2)
    described as ''{relation}'' in (the) DESCRIPTION(1)?
Consider paraphrases but do NOT infer unstated details.

Scoring guide -> 1: absent; 2: weak hint; 3: partial;
4: clear; 5: explicit & unambiguous.

Respond ONLY with an integer 1-5.
```

---

## A.2 DOCENT

### A.2.1 GENERATIONS

We produce generations from the following models:

1. `llava-v1.6-mistral-7b-hf` on HuggingFace (Liu et al., 2024)
2. `Molmo-7B-D-0924` on HuggingFace (Deitke et al., 2024)
3. `gpt-4o-2024-08-06`, accessed on 1/31/25
4. `claude-3-5-sonnet-20241022`, accessed on 1/31/25

We use the same prompt (included below). For `LLaVA-1.5-7B` and `Molmo-D-7B`, we use nucleus sampling Holtzman et al. (2019) with $p = 0.9$ and a temperature of 0.7.

---

**Detailed Description Prompt**

[IMAGE]

Generate a detailed description of this painting, avoiding interpretation and focusing on only its visual elements.

---

### A.2.2 AVOIDING DOUBLE PENALTIES

In Kasai et al. (2022), after identifying an error in precision, the authors correct the error before annotating recall. This avoids doubly penalizing a description for errors in specificity which would unfairly favor more generic descriptions (which are only penalized once, for recall). We instruct our annotators to do the same.

Due to the length of the generations and descriptions in DOCENT, please consult our codebase for example judgments: `https://github.com/amith-ananthram/posh/tree/main/docent/examples/granular`

**Granular Evaluation of Image Descriptions**

Hello, thanks for being part of our research study. Our goal is to accurately characterize the performance of vision-language models (i.e., AI systems that can describe images). By doing so, you'll help us gauge how well such systems would perform in consequential settings such as the automatic generation of accessibility text for people who are blind or have low vision.

In our annotation interface, you'll see 1) an image, 2) a **CORRECT** description of the image and 3) a **GENERATED** description of the image. Your task is to first identify **minimal** spans in the **GENERATED** description that are *mistakes* (e.g. incorrectly added details that are not true of the image; identifications of nouns, their descriptors or their relationships that are not true of the image) and then identify **minimal** spans in the **CORRECT** description that are *missing* (e.g. details not reflected in the **GENERATED** description). For each task, please follow the instructions below:

1) Look at the image. Get a quick sense of any relevant people or objects, their actions and their broader setting.

2) Read the **CORRECT** description of the image.

3) Read the **GENERATED** description of the image.

4) Read the **GENERATED** description of the image again. As you encounter *mistakes* (e.g., incorrectly added details that are not true of the image or nouns, their descriptors or their

Figure 4: The beginning of our granular annotation instructions, also hosted on our GitHub.

**Evaluation of Image Descriptions**

Hello, thanks for being part of our research study. Our goal is to accurately characterize the performance of vision-language models (i.e., AI systems that can describe images). By doing so, you'll help us gauge how well such systems would perform in consequential settings such as the automatic generation of accessibility text for people who are blind or have low vision.

In our annotation interface, you'll see 1) an image, 2) a **CORRECT** description of the image and 3) two **GENERATED** descriptions of the image. Your task is to provide **relative grades** of the **GENERATED** descriptions across *three dimensions*: **mistakes, missing details** and **overall quality**.

**Mistakes** in **GENERATED** descriptions are incorrectly added details or identifications of nouns, their descriptors or their relationships that are not true of the image. **Missing details** are details in the **CORRECT** description that are not accounted for in the **GENERATED** descriptions **after correcting their mistakes**. **Overall quality** is more subjective – we want you to grade the generations by which one is the best stand-in for the **CORRECT** description.

For each task, please follow the instructions below:

1) Look at the image. Get a quick sense of any relevant people or objects, their actions and their broader setting.

Figure 5: The beginning of our coarse annotation instructions, also hosted on our GitHub.

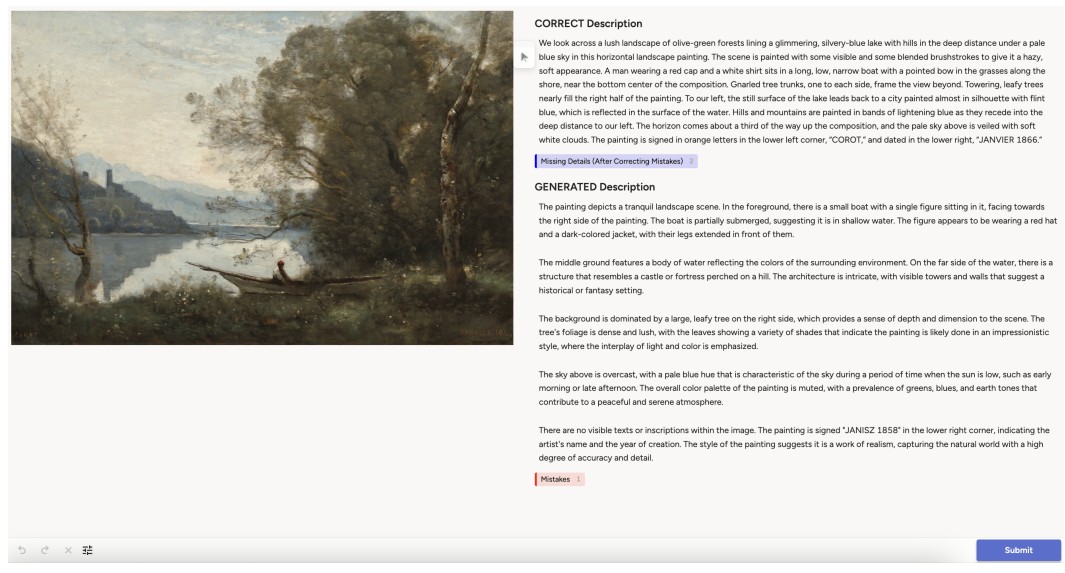

Figure 6: Our granular annotation interface, hosted on Label Studio (https://labelstud.io).

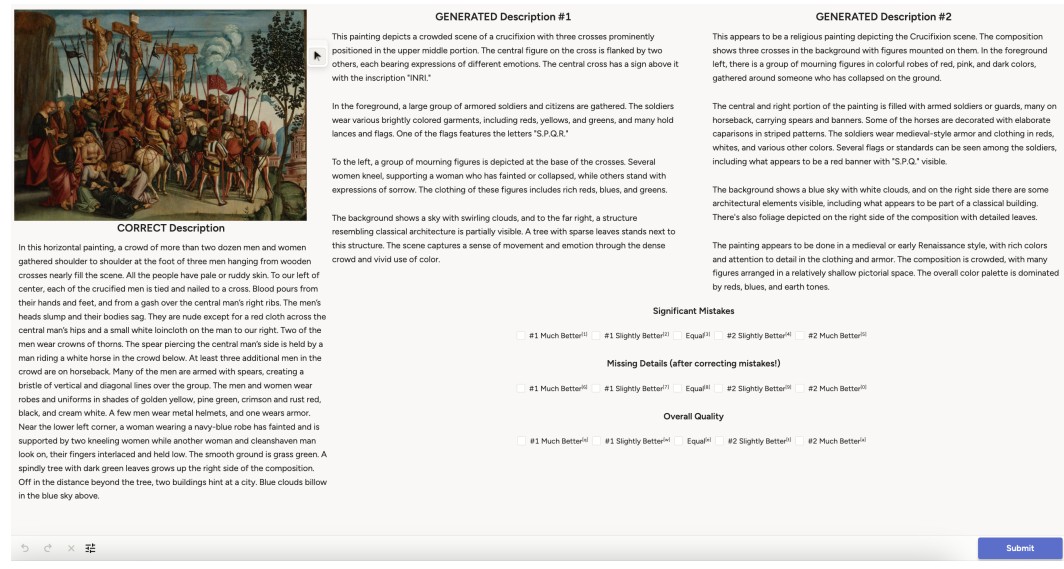

Figure 7: Our coarse annotation interface, hosted on Label Studio (https://labelstud.io).

### A.2.3  DOCENT: AGREEMENT AND JUDGMENT SUMMARY

Table 4: Granular judgments of mistakes and omissions in DOCENT. Left: inter-annotator agreement (relaxed F1, using overlap thresholds of $\geq 1$ token and $\geq 50\%$ of tokens); additionally, the recall $R$ of our expert annotations. Our student judgments exhibit strong inter-annotator agreement and good coverage of our more sparing (see Fig. 8) error annotations. Right: the average percentage of tokens *not* marked as mistakes/omissions for each model (higher is better).

| | student | | expert | | | | llava | molmo | claude | gpt4o |
| | F1 | F1@50 | R | F1 | R@50 | F1@50 | | | | |
|---|---|---|---|---|---|---|---|---|---|---|
| **mistakes** | 0.980 | 0.604 | 1.000 | 0.890 | 0.652 | 0.250 | 0.886 | 0.920 | 0.961 | 0.957 |
| **omissions** | 1.000 | 0.754 | 1.000 | 1.000 | 0.927 | 0.475 | 0.359 | 0.468 | 0.462 | 0.501 |

Table 5: Coarse judgments of precision, recall and overall quality in DOCENT. Top: inter-annotator agreement (Krippendorff $\alpha$ for student, Pearson $\rho$ and average difference for expert). Bottom: the average relative score of each row model compared to each column model (1 indicates the row is much better, 5, the column is much better).

| | mistakes | | | omissions | | | overall quality | | |
|---|---|---|---|---|---|---|---|---|---|
| **student** ($\alpha$) | | 0.509 | | | 0.409 | | | 0.459 | |
| **expert** ($\rho, \Delta$) | | 0.727, 0.633 | | | 0.501, 0.644 | | | 0.492, 0.788 | |
| | **llava** | **molmo** | **claude** | **llava** | **molmo** | **claude** | **llava** | **molmo** | **claude** |
| **molmo** | 2.42 | | | 2.32 | | | 2.14 | | |
| **claude** | 1.92 | 2.37 | | 2.16 | 2.74 | | 1.87 | 2.5 | |
| **gpt4o** | 1.86 | 2.3 | 3.0 | 2.01 | 2.54 | 2.61 | 1.68 | 2.21 | 2.63 |

### A.2.4  DOCENT: GRANULAR AGREEMENT DETAILS

We additionally calculate granular agreement using a more conservative threshold ($\geq 50\%$ token overlap). Here, relaxed F1 remains strong among our art history student annotators (0.612 for mistakes, 0.773 for omissions). Though we observe drops in relaxed F1 when compared to our expert, it is driven by two factors: annotation style, with our expert favoring sparsity, and a relative strictness on the part of our student annotators. This is reflected in the expert annotation recall values in Table 4 where a majority of the spans identified by our expert were also marked by our student annotations for both mistakes (0.652) and omissions (0.927). Thus, our expert annotations are a subset of our stricter student annotations. We provide a side-by-side example of a student annotation and an expert annotation in Fig. 8.

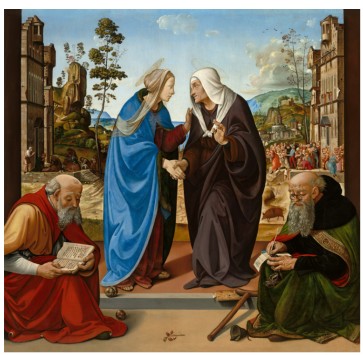

Figure 8: A comparison of our student and expert judgments of omissions for the same generation. Most differences are due to 1) students preferring the specificity of terms like "women" over "figures" and 2) students annotating all the attributes and relations of entities marked as missing, e.g., "skin", "halos", "noses", and the span beginning "painted with a scene...". Generally, expert judgments are a subset of our student judgments for these reasons.

### A.3 EVALUATION

#### A.3.1 METRICS

**Spearman's Rank Correlation Coefficient ($\rho$)**  assesses the monotonic relationship by calculating Pearson's correlation on the ranks of two continuous variables rather than their raw values. It ranges from $-1$ to $+1$, with $+1$ indicating perfect monotonic increasing relationship and $-1$ indicating perfect monotonic decreasing relationship. It's less sensitive to outliers than Pearson's and can detect monotonic non-linear relationships. As the coarse annotations in DOCENT specify the rank of two generated image descriptions, Spearman is well suited for evaluating

**Kendall's $\tau$**  measures the ordinal association between two variables based on the ranks of the data. It ranges from $-1$ to $+1$, where $+1$ indicates perfect agreement between the two rankings, $0$ indicates no association, and $-1$ indicates perfect disagreement. Unlike Pearson's, Kendall's tau is non-parametric and robust to outliers, making it appropriate for non-linear relationships and non-normally distributed data.

#### A.3.2 COARSE SCORE SCALED EVALUATION

We convert each coarse judgment of a generation pair (text$_1$, text$_2$, label) in DOCENT to a numerical score $s$ that reflects the relative rank of text$_1$ and text$_2$. If text$_1$ was marked as `much better` than text$_2$, $s = 2$; `slightly better` than text$_2$, $s = 1$ and `equal` to text$_2$, $s = 0$. Similarly, if text$_2$ was marked as `slightly better` than text$_1$, $s = -1$ and $s = -2$ if `much better` than text$_1$. These numerical scores reflect the relative rank of text$_1$ and text$_2$ and allow us to evaluate the correlation of different metrics $m$ with the coarse judgments in DOCENT by comparing $s$ to $m(\text{text}_1) - m(\text{text}_2)$ with appropriate measures of monotonicity like Spearman's rank correlation $\rho$.

#### A.3.3 GRANULAR BASELINES

**4GramEmbed**  We extract all of the 4-grams from each sentence of a generation and its reference, embed them using `Qwen/Qwen3-Embedding-8B` (Reimers & Gurevych, 2019; Yang et al., 2025) and then calculate the maximum pairwise similarities between generation 4-grams and reference 4-grams. Generation text spans and reference text spans with maximum pairwise similarity scores lower than $0.7$ were predicted as mistakes and omissions respectively, a threshold chosen to maximize the macro F1 scores reported for **4GramEmbed** in Table 2.

**SGEmbed**  We extract all of the components (objects, attribute-object pairs, and object-relation-object triples) from the scene graphs of a generation and its reference extracted for POSH in Section 3, embed them using `Qwen/Qwen3-Embedding-8B` (Reimers & Gurevych, 2019; Yang et al., 2025) and then calculate the maximum pairwise similarities between the generation components and the reference components. Generation components and reference components with maximum pairwise similarity scores lower than $0.8$ were predicted as mistakes and omissions respectively, a threshold chosen to maximize the macro F1 scores reported for **SGEmbed** in Table 2.

#### A.3.4 COARSE BASELINES

When prompting GPT4o and GPT5[14] to evaluate our generated detailed image descriptions, we use three different prompts depending on whether we are including the image (reference free) or including the reference. Additionally, we experiment with a more complicated prompt that includes a detailed scoring rubric for each score type (mistakes, omissions and overall quality) though we find that this setting underperforms the simpler prompts below.

---

[14] `gpt-4o-2024-08-06` and `gpt-5-2025-08-07` (with minimal reasoning) accessed on 9/17/2025

---

**Image Only**

[IMAGE]

Generated Detailed Description: [GENERATION]

Please provide numerical scores (from 0 to 5) for the precision (e.g. mistakes in the generated description), recall (e.g. missing details from the image), and overall quality of the generated detailed description compared to the image. Output your answer as a JSON dictionary with the keys 'precision', 'recall', and 'overall_quality'.

---

**Reference Only**

Reference Detailed Description: [REFERENCE]

Generated Detailed Description: [GENERATION]

Please provide numerical scores (from 0 to 5) for the precision (e.g. mistakes in the generated description), recall (e.g. missing details from the reference description), and overall quality of the generated description compared to the reference description. Output your answer as a JSON dictionary with the keys 'precision', 'recall', and 'overall_quality'.

---

**Image & Reference**

[IMAGE]

Reference Detailed Description: [REFERENCE]

Generated Detailed Description: [GENERATION]

Please provide numerical scores (from 0 to 5) for the precision (e.g. mistakes in the generated description), recall (e.g. missing details from the reference description), and overall quality of the generated detailed description compared to the image and the reference description. Output your answer as a JSON dictionary with the keys 'precision', 'recall', and 'overall_quality'.

---

### A.3.5 REINFORCEMENT LEARNING

We train `Qwen2.5-VL-7B` on the $1,000$ images in DOCENT's training set in two settings:

1. supervised fine-tuning (SFT) with full parameter updates using a learning rate of $1e-5$, a linear warmup ratio of $0.1$, and an effective batch size of $64$ for 5 epochs, choosing the checkpoint with the lowest loss on DOCENT's validation set

2. DAPO (Yu et al., 2025) with full parameter updates, implemented with TRL (von Werra et al., 2020), using a learning rate of $1e-6$, 20 warmup steps, 8 generations per sample (with a temperature of $1.0$ and $top_p = 0.7$), $\epsilon = 0.28$, $\beta = 0$, and an effective batch size of $64$ for a single epoch, choosing the final checkpoint

We ask seven graduate students in NLP to compare and evaluate our SFT and DAPO generations (greedily sampled) for 40 images from DOCENT's test set. Additionally, we collect three annotations for five of these images to calculate agreement.

## A.4 RESULTS

### A.4.1 COARSE

Table 6: All coarse metrics evaluated on DOCENT and CapArena, identified with Θ (parameter count, in billions), 📄 (requires a reference), 🖼 (requires an image) and 🔁 (replicable). "acc" indicates accuracy at predicting the better generation (or "tie") in each judged pair. For DOCENT, $\rho$ / $\tau$ indicate the Spearman rank / Kendall's $\tau$ correlations between differences in the metric and differences in the rank of the generations in each pair. For CapArena, $\rho$ / $\tau$ indicate the Spearman rank / Kendall's $\tau$ correlations between model ELO rankings derived from metric scores and human judgments. **Bold** indicates the best replicable metric while underlining indicates the best metric overall. Gray cells indicate correlations that are *not* statistically significant at $\alpha = 0.05$. POSH beats all replicable baselines and GPT4o on DOCENT in all settings (mistakes, omissions and overall quality) while remaining perfectly replicable. Moreover, POSH is robust, achieving the second best score among replicable metrics on CapArena.

| | | | | | DOCENT | | | | | | | | | CapArena | | |
| | | | | | Mistakes | | | Omissions | | | Overall Quality | | | Desc | Model | |
| | Θ | 📄 | 🖼 | 🔁 | acc | $\rho$ | $\tau$ | acc | $\rho$ | $\tau$ | acc | $\rho$ | $\tau$ | acc | $\rho$ | $\tau$ |
|---|---|---|---|---|---|---|---|---|---|---|---|---|---|---|---|---|
| length | | | | ✓ | 30.5 | −0.270 | −0.206 | 37.8 | −0.002 | −0.001 | 38.0 | −0.160 | −0.121 | 58.7 | 0.710 | 0.582 |
| BLEU-4 | | ✓ | | ✓ | 34.2 | −0.070 | −0.053 | 42.5 | 0.118 | 0.087 | 42.8 | 0.051 | 0.038 | 47.4 | 0.424 | 0.319 |
| CIDER | | ✓ | | ✓ | 32.0 | −0.118 | −0.089 | 37.5 | −0.009 | −0.007 | 37.8 | −0.106 | −0.079 | 38.4 | −0.279 | −0.209 |
| METEOR | | ✓ | | ✓ | 36.0 | −0.103 | −0.078 | 46.2 | 0.260 | 0.197 | 44.8 | 0.113 | 0.084 | 57.6 | 0.785 | 0.582 |
| ROUGE-LS | | ✓ | | ✓ | 37.5 | 0.251 | 0.190 | 44.0 | 0.214 | 0.161 | 47.3 | 0.210 | 0.158 | 45.8 | 0.180 | 0.199 |
| SPICE | | ✓ | | ✓ | 41.3 | 0.308 | 0.234 | 55.0 | 0.464 | 0.360 | 58.5 | 0.458 | 0.349 | 41.7 | 0.275 | 0.231 |
| CAPTURE | | ✓ | | ✓ | 43.3 | 0.259 | 0.194 | 53.8 | 0.447 | 0.340 | 56.0 | 0.453 | 0.347 | 52.5 | 0.613 | 0.538 |
| CLIPScore | | | ✓ | ✓ | 45.3 | 0.145 | 0.108 | 47.0 | 0.176 | 0.133 | 53.5 | 0.181 | 0.136 | 32.5 | −0.574 | −0.451 |
| FLEUR | 13 | | ✓ | ✓ | 35.2 | −0.053 | −0.040 | 38.5 | 0.029 | 0.020 | 41.2 | −0.040 | −0.031 | 45.8 | 0.393 | 0.297 |
| Prometheus | 8x7 | ✓ | | ✓ | 51.2 | 0.014 | 0.011 | 49.8 | 0.136 | 0.116 | 58.5 | −0.007 | −0.007 | 53.9 | 0.859 | 0.648 |
| Qwen3 | 32 | ✓ | | ✓ | 57.7 | 0.282 | 0.235 | 53.5 | 0.286 | 0.253 | 61.2 | 0.289 | 0.257 | 56.2 | 0.899 | 0.714 |
| LLaVa Critic | 72 | ✓ | ✓ | ✓ | **62.8** | 0.412 | 0.351 | 57.0 | 0.509 | 0.430 | 66.8 | 0.546 | 0.461 | **64.0** | **0.987** | **0.934** |
| DSG | | | ✓ | | 41.3 | 0.091 | 0.068 | 37.3 | 0.033 | 0.024 | 45.3 | 0.017 | 0.012 | - | - | - |
| DCScore | | ✓ | ✓ | | 62.8 | 0.541 | 0.422 | 54.0 | 0.395 | 0.298 | 62.8 | 0.471 | 0.362 | - | - | - |
| GPT4o | | | ✓ | | 63.2 | 0.469 | 0.400 | 55.5 | 0.338 | 0.274 | 66.7 | 0.477 | 0.393 | 53.6 | 0.868 | 0.692 |
| GPT4o | | ✓ | | | 53.3 | 0.324 | 0.261 | 50.0 | 0.277 | 0.215 | 60.8 | 0.388 | 0.297 | 56.7 | 0.867 | 0.685 |
| GPT4o | | ✓ | ✓ | | 58.5 | 0.484 | 0.396 | 56.0 | 0.380 | 0.303 | 67.3 | 0.510 | 0.402 | 55.4 | 0.890 | 0.802 |
| GPT5 | | | ✓ | | 66.0 | 0.584 | 0.476 | 55.5 | 0.454 | 0.345 | 69.2 | 0.593 | 0.466 | 56.9 | 0.916 | 0.802 |
| GPT5 | | ✓ | | | 62.5 | 0.511 | 0.423 | 53.2 | 0.421 | 0.332 | 68.0 | 0.540 | 0.440 | 59.1 | 0.956 | 0.846 |
| GPT5 | | ✓ | ✓ | | 68.2 | 0.604 | 0.494 | 56.3 | 0.477 | 0.366 | 67.2 | 0.602 | 0.475 | 62.1 | 0.934 | 0.846 |
| POSH | 14 | ✓ | | ✓ | 60.7 | **0.519** | **0.405** | **62.7** | **0.581** | **0.451** | **70.7** | **0.599** | **0.466** | 59.2 | 0.931 | 0.796 |

Table 7: Annotator agreement (Krippendorff's $\alpha$) and aggregate preferences between variants of Qwen2.5-VL-7B trained on DOCENT: tuned with DAPO using POSH ("POSH-tuned") and supervised fine-tuned ("SFT"). A POSH-tuned generation earns a score between $-2$ and $2$ based on how much worse or better it is than its SFT counterpart. Reported numbers are averages of these scores.

| | Mistakes | Omissions | Overall Quality |
|---|---|---|---|
| $\alpha$ | 0.235 | 0.464 | 0.184 |
| POSH-tuned vs SFT | −0.243 | 0.432 | 0.135 |

### A.4.2 SUBCOMPONENT PERFORMANCE

To evaluate the scene graph extraction subcomponent of POSH, we hand annotate scene graphs for ten descriptions in DOCENT, five references and five generations. We measure precision, recall and F1 for entities, attributes and relations and, for each matched element pair (i.e. entity, attribute or relation), accuracies for coreference resolution, attribute attachment and relation head and tail attachment. The numbers reported in Table 8 are averaged across the ten descriptions.

To evaluate the scene graph element verification subcomponent of POSH, we manually answer 620 templated questions for two randomly sampled generation-reference pairs in DOCENT and compare them to POSH's presence scores. The numbers reported in Table 9 are the maximum achievable F1 scores across all alerting thresholds. The errors in this subcomponent stem from generations specifying correct details not present in the reference, relative permissiveness toward interpretive language in generations and limitations of the POSH's templating and unique identifier discovery logic. Nevertheless, the scores speak to the strength of the POSH framework and the potential of future subcomponent improvements to yield further gains in performance.

Table 8: Evaluation of POSH's extracted scene graphs for ten descriptions in DOCENT. "P" indicates precision, "R" indicates recall, "coref" indicates accuracy at predicting each matched entity's first mention, "head" indicates accuracy at predicting each matched attribute or relation's entity subject and "tail" indicates accuracy at predicting each matched relation's entity object.

| Entities | | | | Attributes | | | | Relations | | | | |
|---|---|---|---|---|---|---|---|---|---|---|---|---|
| P | R | F1 | coref | P | R | F1 | head | P | R | F1 | head | tail |
| 0.937 | 0.883 | 0.909 | 0.925 | 0.881 | 0.917 | 0.898 | 0.951 | 0.978 | 0.794 | 0.872 | 0.850 | 0.926 |

Table 9: Evaluation of POSH's scene graph verification for 620 elements from two generation-reference pairs in DOCENT, broken out by mistakes (verifying generation scene graph elements in a reference) and omissions (verifying reference scene graph elements in a generation).

| Mistakes F1 | Omissions F1 | Overall F1 |
|---|---|---|
| 0.941 | 0.754 | 0.852 |

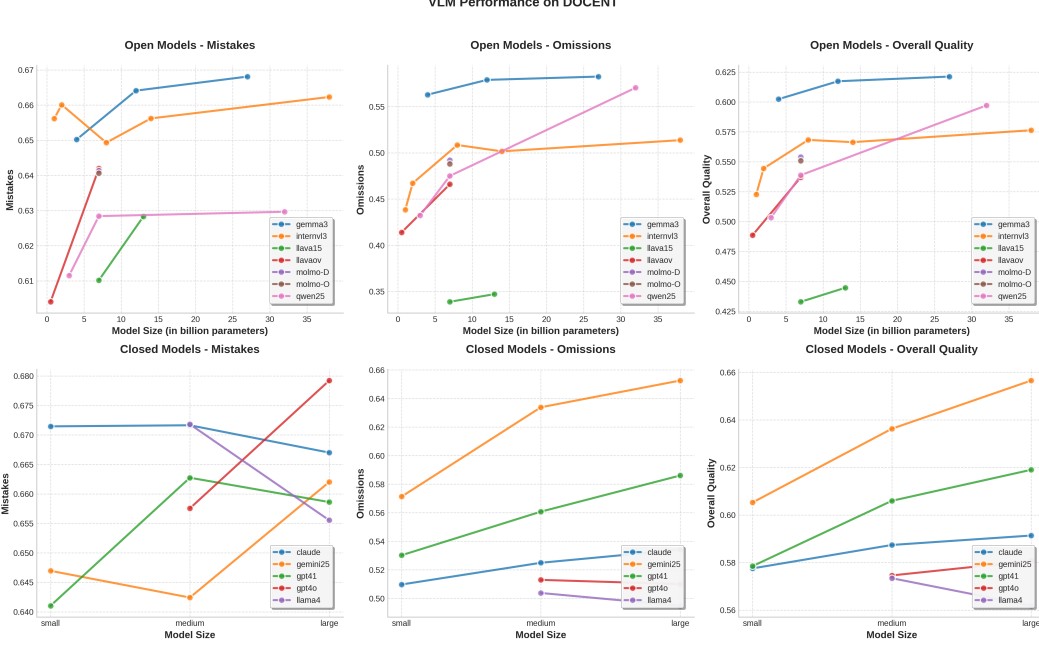

Figure 9: Performance of open and closed VLMs on DOCENT, as measured by POSH. While open models are competitive when it comes to mistakes in their detailed descriptions, they lag behind in their omissions, covering less of DOCENT's reference descriptions than closed models.

