# OpenReview forum: "PoSh: Using Scene Graphs to Guide LLMs-as-a-Judge for Detailed Image Descriptions"
_ICLR.cc/2026/Conference — ICLR 2026 Poster_

### Official Review · Reviewer_XyHZ · 2025-10-29

**Soundness:** 3
**Presentation:** 3
**Contribution:** 2
**Rating:** 4
**Confidence:** 2

**Summary:**

The paper proposes PoSh, a new metric for evaluating fine-grained image descriptions. PoSh extracts scene graphs from both reference and generated descriptions, preserving the object–attribute–relation structure, and uses these graphs as a structured scoring criterion to guide open-source LLMs in performing fine-grained judgments. This enables the identification of mistakes and omissions in generated text.

The authors construct a new benchmark, DOCENT, comprising 1,750 artworks from the National Gallery of Art, expert-written descriptions, outputs from VLMs, 300 fine-grained annotations, and 600 coarse-grained pairwise ratings provided by art history students. Experiments show that PoSh outperforms existing metrics on both DOCENT and CapArena. It can also serve effectively as a reward function in reinforcement learning.

**Strengths:**

1. PoSh addresses a clear gap in evaluating detailed image descriptions. By using scene graphs as structured rubrics, PoSh enables error localization and produces human-aligned, interpretable scores.
2. Built entirely on open-weight models and public tools, PoSh avoids reliance on proprietary APIs (e.g., GPT-4o), making it accessible and deployable for researchers with limited resources.
3. The paper introduces DOCENT—a novel dataset of expert-written art descriptions paired with granular and coarse human judgments from domain-knowledgeable annotators. This enables evaluation for complex visual domains.

**Weaknesses:**

1. **Computational overhead may hinder scalability**: The pipeline, comprising scene graph extraction, multi-pass identifier generation, and LLM-based QA, is considerably more complex than standard metrics. The paper does not include ablation studies or timing analyses of individual components. Without such efficiency profiling, it remains unclear whether PoSh offers a favorable trade-off between evaluation quality and computational cost in large-scale settings.
2. **Lack of validation as a data curation tool for MLLM training**: While the paper demonstrates PoSh’s effectiveness as a reinforcement learning reward signal, it does not explore its utility in filtering or ranking training data for MLLM fine-tuning. Comparing models trained on PoSh-filtered data versus those trained with other metrics would better establish PoSh’s practical value in the full model development lifecycle.

**Questions:**

Do captioning models that incorporate structural priors (e.g., spatial or scene graph inputs) benefit disproportionately under PoSh?
Some recent methods explicitly inject visual relational priors—such as object positions or scene graph structures—into the captioning process.
1. Since PoSh itself uses scene graphs as evaluation rubrics, such models might receive inflated scores due to structural alignment between generation and evaluation, rather than superior semantic fidelity. Have the authors evaluated PoSh on outputs from graph-aware captioning systems (e.g., SG-LLaVA [1])?
2. I also wonder whether the authors think that incorporating structural priors during generation might be more important than adding them during verification.

[1] Jingyi Wang, Jianzhong Ju, Jian Luan, and Zhidong Deng. LLaVA-SG: Leveraging Scene Graphs as Visual Semantic Expression in Vision-Language Models.

---

> ### Author Response · Authors · 2025-11-25
> **Response to Reviewer XyHZ**
>
> We’d like to thank the reviewer for their thoughtful engagement with our work.  We appreciate that they found that our work “**addresses a clear gap in evaluating detailed image descriptions**”, is “**accessible and deployable by researchers with limited resources**”, and that our dataset “**enables evaluation for complex visual domains**”.
>
> Below, we address the questions raised by the reviewer.
>
> > ### Computational overhead
>
> PoSh achieves its thorough evaluation of detailed image descriptions through an implementation optimized for computational efficiency.  Scene graph extraction is extremely fast, relying on BERT-sized models for dependency parsing and coreference resolution.  Though granular scoring involves the evaluation of a large number of questions for each generation-reference pair, PoSh does so through vLLM, leveraging optimizations like continuous batching and prefix caching to speed up generation.  Prefix caching in particular allows PoSh to evaluate all candidate identifiers together (line 240) extremely efficiently (as they all share the long generation/reference text as a prefix).
>
> Through these optimizations, on a single H100 GPU, **PoSh scores the 400 generated descriptions in DOCENT in 15 minutes, 38 seconds (one description every 2 seconds).  In contrast, DCScore, which relies on GPT4 to extract and verify factoids, takes 2 hours, 47 minutes and 21 seconds  (one description every 25 seconds)**.
>
> Finally, we would like to note that as most metrics are poor proxies for the human judgments in DOCENT, the alternative to using PoSh is **manual evaluation, which takes 18 minutes for each generation-reference pair** (see Table 1, line 160).
>
> We have updated our manuscript to include this runtime explanation, analysis and comparison (lines 483 - 487).
>
> > ### Use of PoSh as a data curation tool for MLLM training
>
> **As PoSh is a reference based metric, it cannot be used for curating data.**  We focus on reference-based evaluation as image description is underspecified (lines 124 - 126).  Downstream tasks have context-specific requirements that can only be evaluated with references (for example, detailed assistive text will look different than detailed descriptions focused on facial expressions [1]).
>
> [1]  “Context Matters for Image Descriptions for Accessibility: Challenges for Referenceless Evaluation Metrics” (Kreiss et al., EMNLP 2022): https://arxiv.org/abs/2205.10646
>
> > ### Captioning models with structural priors
>
> While we would like to evaluate PoSh on generations from models that incorporate structural priors, we have been unable to find any such models where the weights / code are publicly available.  **While the SG-LLaVa paper the reviewer links to is certainly interesting, they have not published a codebase.**
>
> Generally speaking, efforts that use structural priors to guide generation extract and describe structure from images.  In contrast, PoSh extracts and validates structure from text.  Thus, if the contents of those two structured representations differ (the image structure and the reference text structure), PoSh will correctly penalize these approaches.
>
> Nevertheless, we have called this out in the limitations section of our updated manuscript as a possible concern (lines 509 - 512).
>
> > ### Structural priors in generation vs evaluation
>
> This is an interesting question and while we do not presume to know the answer, we suspect that structural priors might be more useful for verification than generation.  To date, the bitter lesson has shown that naive scaling yields significant gains in model performance.  In contrast, evaluation has specific requirements that favor structured approaches – in particular, being able to ground a coarse score in granular scores (interpretability).  Structured approaches like scene graphs are a principled way for generating and applying example-specific rubrics to guide evaluation.
>
> Thank you for requesting runtime analysis and a call out about structural priors in evaluated generations.  We have updated our manuscript to address both and it is stronger for their inclusion.

---

### Official Review · Reviewer_cNnP · 2025-11-01

**Soundness:** 3
**Presentation:** 3
**Contribution:** 3
**Rating:** 6
**Confidence:** 4

**Summary:**

This paper presents POSH, a new metric for detailed image description. It computes scores grounded in fine-grained errors by adopting scene graphs to guide LLMs-as-Judge. POSH is interpretable and aligns better with human validation. Additionally, they propose DOCENT, a new dataset of artwork, references, and descriptions. They validate POSH on DOCENT and find it has better correlation with human judgments. By benchmarking open and closed-source models on DOCENT, they identify strengths and weaknesses of VLMs in understanding images.

**Strengths:**

- They focus on an important aspect of VLM understandings. They focus on the detailed description, and the metric is interpretable.
- They propose a benchmark with expert-written descriptions and 900 granular & coarse judgments from raters. The manual effort is massive.
- They open-sourced the benchmark and metric, which will benefit the community.
- They evaluate multiple open-source and closed-source models.

**Weaknesses:**

- The writing could be better. For example, in the table, they use POSH to denote the finetuned Qwen model with POSH reward, while POSH is a metric in the meantime. This is a bit confusing.
- For the findings of POSH as a reward function, they only experiment with the Qwen2.5-VL-7B model. The findings may not be model-agnostic.
- It is concerning that POSH works better on their proposed DOCENT benchmark but is adequate on other benchmarks like CapArena.

**Questions:**

- Could you provide results with other VLMs other than Qwen2.5-VL?
- What is the text and image distribution difference between DOCENT and CapArena? Is there any other caption benchmarks you could use to validate the effectiveness of POSH?

---

> ### Author Response · Authors · 2025-11-25
> **Response to Reviewer cNnP**
>
> We’d like to thank the reviewer for their thoughtful engagement with our work.  We appreciate that they found the task we focus on to be “**important**”, our emphasis on interpretability to be valuable, our dataset to reflect “**massive manual effort**”, our “**open-source metric and dataset to be contributions that benefit the community**”, and our evaluation of our metric to be thorough.
>
> Below, we address the questions raised by the reviewer.
>
> > ### Confusions in writing
>
> Thank you for drawing our attention to this table.  We have **updated the text to make this clearer (lines 469, 1277 - 1285)**.
>
> > ### PoSh as a reward function
>
> While we would like to have evaluated PoSh as a reward function on additional models, unfortunately, doing so is quite compute intensive, requiring full parameter updates on a 7B parameter model.  Though we attempted to expand these results, we were unable to get access to the compute required for doing so (and the subsequent human judgments necessary for validating their quality) before the end of the rebuttal period.
>
> > ### DOCENT vs CapArena
>
> The art in DOCENT exhibits more visual complexity than the natural images in CapArena, which were personal photographs from one of the authors of DOCCI.  For example, one of the images in DOCCI is of a plain clock.   The greater complexity of DOCENT shows up in the numbers.  **Compared to the natural images in CapArena, the references in DOCENT are more than twice as long and have scene graphs that are nearly three times as large (Table 1, line 160 vs 159); as a percentage, twice as many of its images contain two or more objects and 10 times as many of its images contain two or more people (line 372-373).**   Thus, DOCENT is a useful testbed for tracking progress in VLMs.
>
> > ### Validating PoSh on CapArena
>
> PoSh is the **most performant open-source metric (outperforming GPT4o-as-a-Judge) on DOCENT and the second most performant open-source metric on CapArena** – it is edged out by the much larger LLaVaCritic-72B.  Nevertheless, we believe its strong scores here speak to its robustness to image type.
>
> In particular, PoSh is effective at detailed description of complex imagery.  It performs better on DOCENT, which, as described above, exhibits considerably more visual complexity than CapArena.  Moreover, **on subsets of CapArena that are more visually complex, like photographs of multiple people that require individual description, PoSh outperforms LLaVaCritic-72B (lines 462-464)**.  Thus, **PoSh is better tailored to evaluating detailed descriptions of complicated imagery** which are more relevant to future progress in VLMs.
>
> While we would have liked to have evaluated PoSh against more judgments of generated detailed image descriptions, unfortunately, beyond DOCENT and CapArena, no such judgments have been published (see Table 1).  This speaks to the value of DOCENT in expanding detailed image description to a new domain.
>
> Thank you for suggesting improvements to the language in our manuscript and for asking for further explanation of the strengths of PoSh on complex imagery.  We have updated our manuscript to address your questions and believe it is stronger as a result.

---

### Official Review · Reviewer_hCTY · 2025-11-01

**Soundness:** 3
**Presentation:** 4
**Contribution:** 4
**Rating:** 8
**Confidence:** 4

**Summary:**

The authors propose PoSh, a reference-based metric for long-form image captioning. PoSh works by first constructing scene graphs for both the reference and generated captions. The metric then constructs templated questions for both the reference and generated captions' scene graphs and answers these questions with an LLM judge to determine the recall and precision, respectively, of the generated caption. As the metric generates numerous questions for both scene graphs, PoSh can yield both granular and coarse assessments of caption quality.

The authors validate alignment with human judgements on CapArena, a pre-existing benchmark, and DOCENT, a dataset containing artwork with expert-written captions that they introduce. PoSh outperforms baseline metrics for granular identification of mistakes and omissions and is either competitive with or outperforms existing metrics for coarser evaluations. The authors also validate their metric as a reward function for RL training, finding improvement over simple SFT.

**Strengths:**

The paper has numerous strengths:
- Firstly, the task of long-form image captioning evaluation is an important one as models continuously improve in capabilities. PoSh acts as an important contribution within this space by proposing a straightforward reference-based metric that converts the references and generations into scene graphs and using these to assess precision and recall. Particularly, the use of questions to assess both precision and recall lends the metric interpretability and granularity.
- The method outperforms prior baselines for granular evaluations and is competitive with or better than other metrics for coarse evaluations.
- The authors propose a new benchmark, DOCENT, with the novel domain of visual art. DOCENT is coupled with expert-written reference captions and human judgements for a range of vision-language models. This not only helps the evaluation of the metric but could also be used as a testbed for future metrics or for the generation quality of other vision-language models.
- Additionally evaluating PoSh's performance as a reward model makes the evaluation of the metric very complete. I can imagine the granular nature of PoSh being also used to generate targeted natural language feedback for models.

**Weaknesses:**

The main weaknesses I can see are:
- PoSh is going to be sensitive to the accuracy of the extracted scene graphs, where there could be errors either during the dependency parsing process or during coreference resolution. Figure 3 marking "painting" as a mistake acts as one example of this.
- While I think PoSh could act as a strong reward model, it does presume access to detailed reference captions, which is expensive to curate on a large scale. PoSh being reference-based similarly restricts its use for evaluation to dedicated datasets for this purpose.

**Questions:**

The metric's performance on the various evaluations already gives good signal regarding its quality. The paper would nonetheless be improved through an evaluation of the intermediate components of the metric itself. For instance, how accurate is the scene graph extraction (as measured via precision and recall against reference human-annotated scene graphs)? Alternatively, as this might be more feasible during the rebuttal period, how accurate are the LLM judge's answers?

---

> ### Author Response · Authors · 2025-11-25
> **Response to Reviewer hCTY**
>
> We’d like to thank the reviewer for their thoughtful engagement with our work.  We appreciated that they found our paper had “**numerous strengths**” including making an “**important contribution**” to the “**important [task] of long-form image captioning**”, our emphasis on interpretability and granularity to be well-placed, our evaluation of PoSh to be “**very complete**” (including its value as a reward function) and our introduction of DOCENT to be a useful testbed for evaluating future metrics / the generation quality of future VLMs in a “**novel domain**”.
>
> Below, we address the questions raised by the reviewer.
>
> > ### Sensitivity of PoSh to extracted scene graphs
>
> It is certainly true that model based scene graph extraction may introduce errors.  We’d like to note that PoSh can be thought of as a framework which can easily benefit from future improvements to scene graph extraction through simple replacement of the corresponding module.
>
> Nevertheless, we have manually labeled 10 scene graphs to measure PoSh’s scene graph quality directly.  This involved annotating each entity, its coreferences, its attributes and its relations in descriptions up to 20 sentences long and took ~45 minutes for each text.  We evaluated these gold quality scene graphs against those extracted by PoSh’s scene graph sub component, finding that **PoSh’s scene graphs achieve an average F1 of 0.892**:
>
> - Entity Precision: 0.937
> - Entity Recall: 0.883
> - Entity F1: 0.909
> - Coref Accuracy: 0.925
>
> - Attribute Precision: 0.881
> - Attribute Recall: 0.917
> - Attribute F1: 0.898
> - Attribute Head Accuracy: 0.951
>
> - Relation Precision: 0.978
> - Relation Recall: 0.794
> - Relation F1: 0.872
> - Relation Head Accuracy: 0.850
> - Relation Tail Accuracy: 0.926
>
> These scores speak to the quality of PoSh’s extracted scene graphs which offer complete coverage over all of the visual details specified in a generated description and its accompanying reference description.  The strength of these scene graphs is also reflected when evaluating PoSh against human judgments (Table 2 and Table 3), as error filled scene graphs would degrade its performance relative to its peers.
>
> We have updated our manuscript to include this additional human evaluation (lines 476 - 480, 1287 - 1293, 1303 - 1310).
>
> > ### Performance of LLM-as-a-Judge’s QA answers
>
> In addition to hand annotating scene graphs, we have hand annotated 620 scene graph elements to evaluate the performance of our LLM’s verification of their presence.  Across all of our alerting thresholds, **our verification component achieves strong alignment with our labels (F1 of 0.852)**, speaking to the strength of PoSh’s subcomponents.  We have updated our manuscript to include this additional human evaluation with an analysis of its underlying errors (lines 476 - 481, 1294 - 1300, 1312-1318)
>
> - Mistakes F1 = 0.941
> - Omissions F1 = 0.754
> - Overall f1 = 0.852
>
> > ### PoSh as a reward model
>
> The reviewer is right to call out that using PoSh as a reward model presumes access to detailed reference captions which may be expensive to curate on a large scale.  While this is certainly true, we’d like to draw the reviewer’s attention to contemporary efforts for generating high quality synthetic detailed image descriptions by relying on compositions of VLMs in tandem [1].  Such data could be freely scaled for use in reinforcement learning with PoSh.  Additionally, as the reviewer kindly noted, PoSh’s granular scores could be a valuable source of token level feedback for a stronger learning signal [2].  We believe that the RL results we present here suggest that the combination of these two approaches could prove powerful.  We have updated our manuscript (lines 472 - 475) to highlight these exciting future directions.
>
> [1] “DenseFusion-1M: Merging Vision Experts for Comprehensive Multimodal Perception” (Li et al., Neurips 2024): https://arxiv.org/abs/2407.08303
> [2] “Preference-grounded Token-level Guidance for Language Model Fine-tuning” (Yang et al., Neurips 2023): https://arxiv.org/abs/2306.00398
>
> Thank you for suggesting a more thorough evaluation of PoSh’s intermediate components and for asking for further explanation of PoSh’s potential as a reward function.  We have updated our manuscript to answer all of your questions and believe our work is stronger for it.

---

### Official Review · Reviewer_6TR7 · 2025-11-01

**Soundness:** 3
**Presentation:** 3
**Contribution:** 3
**Rating:** 4
**Confidence:** 4

**Summary:**

This paper introduces a new metric for evaluating detailed image descriptions, termed POSH. POSH utilizes scene graphs as structured guidelines to direct LLMs in assessing fine-grained errors in image descriptions, such as compositional correctness. POSH offers a replicable and interpretable evaluation experience, outperforming existing metrics, including GPT4o. Additionally, the authors introduce a new dataset, DOCENT, which contains artwork paired with expert-written descriptions and model-generated descriptions. The dataset also includes annotations from experts, providing a challenging benchmark for evaluating the detailed description of images. Furthermore, the authors proof that POSH can be used as a reward function to achieve better performance than standard sft.

**Strengths:**

* The paper is well-written and easy to follow.
* Evaluating the image description is indeed a non-trivial task, and the proposed new metric for evaluating detailed descriptions is important for the field of image captioning.
* I agree that a good metric for image-description should be grounded on fine-grained cues, localized on text spans.
* The paper introduced the DOCENT dataset, which includes expert-written descriptions and annotations, and the quality is well controlled.

**Weaknesses:**

* POSH is reliance on a model to generate the scene graph introduces inaccuracies and errors, which could be a potential bottleneck for its effectiveness.
* The use of scene graphs to evaluate image-text alignment has been discussed in previous papers like [1]; the authors need to clarify the uniqueness of POSH.
* The proposed dataset covers artworks, but in practical applications, images of natural scenes are more common.

[1] Davidsonian Scene Graph: Improving Reliability in Fine-grained Evaluation for Text-to-Image Generation

**Questions:**

* Why choose artworks as a benchmark? Instead of other more common or more representative domains
* How to ensure repeatability, as there are probabilistic models used (qwen3)?

---

> ### Author Response · Authors · 2025-11-25
> **Response to Reviewer 6TR7 (part 1/2)**
>
> We’d like to thank the reviewer for their thoughtful engagement with our work.  We appreciated that they found our paper to be “**well-written**”, our area of focus (evaluation of detailed image description) to be “**non-trivial**”, our proposed “**metric… [to be] important**”, our emphasis on metric grounding to be well-placed and our dataset DOCENT to be of “**quality [that is] well controlled**”.
>
> Below, we address the questions raised by the reviewer.
>
> > ### Comparison to Davidsonian Scene Graph
>
> There are several differences between the Davidsonian Scene Graph (DSG) metric and PoSh.  While the emphasis of DSG is on evaluating text-to-image models, PoSh was designed specifically for evaluating detailed image descriptions and is tailored to the unique challenges it poses.  This is most clearly seen when evaluating DSG against the judgments in DOCENT, where its **accuracies are near chance and its correlations are near zero** (line 1269).
>
> The reasons for this are numerous.  First, the emphasis of DSG is on evaluating text-to-image models: it compares an image to scene graph elements extracted from a prompt text through visual question answering.  As such, it cannot serve as a reference-based metric for evaluating detailed image descriptions.  We highlight the importance of reference-based metrics for the underspecified task of image description in lines 122 - 125: downstream tasks have context-specific requirements that can only be evaluated with references (for example, detailed assistive text will look different than detailed descriptions focused on facial expressions [1]).
>
> Additionally, DSG was designed for image generation prompts at most 3 sentences long (in contrast, PoSh is able to compare generations and references that are 10 - 20 sentences long); DSG prompts GPT4 for atomic propositions that are not localized to text spans (in contrast, PoSh’s coarse scores are grounded in localized granular scores with open models, allowing error visualization, better interpretability and replicability), DSG's atomic propositions do not have special handling for entity collisions, e.g, when there are two women specified in an image prompt (in contrast, PoSh tests discriminating identifiers for colliding entities to allow unique validation of their presence), and finally, DSG applied to detailed image descriptions measures only precision, validating the presence of generation elements in its source image (in contrast, PoSh measures both precision and recall, penalizing generations for omitting important details).
>
> In summary, PoSh was designed specifically for detailed image description evaluation and is constructed in a way to allow easy replication of its scores among researchers and practitioners.  We have included language in the Appendix of our updated manuscript (lines 740-757) to make clear these differences.
>
> [1] “Context Matters for Image Descriptions for Accessibility: Challenges for Referenceless Evaluation Metrics” (Kreiss et al., EMNLP 2022): https://arxiv.org/abs/2205.10646
>
> > ### Reliance on a model to generate scene graphs
>
> While it is certainly true that model based scene graph extraction may introduce errors, we note that without relying on a model, researchers and practitioners would be required to hand annotate their own scene graphs for their detailed image descriptions, a task which takes up to 45 minutes per instance.  Moreover, PoSh can be thought of as a framework which can easily benefit from future improvements to scene graph extraction through simple replacement of the corresponding module.
>
> Nevertheless, we have manually labeled 10 scene graphs to measure PoSh’s scene graph quality directly.  This involved annotating each entity, its coreferences, its attributes and its relations in descriptions up to 20 sentences long.  We evaluated these gold quality scene graphs against those extracted by PoSh’s scene graph sub component, finding that **PoSh’s scene graphs achieve an average F1 of 0.892**:
>
> - Entity Precision: 0.937
> - Entity Recall: 0.883
> - Entity F1: 0.909
> - Coref Accuracy: 0.925
>
> - Attribute Precision: 0.881
> - Attribute Recall: 0.917
> - Attribute F1: 0.898
> - Attribute Head Accuracy: 0.951
>
> - Relation Precision: 0.978
> - Relation Recall: 0.794
> - Relation F1: 0.872
> - Relation Head Accuracy: 0.850
> - Relation Tail Accuracy: 0.926
>
> These scores speak to the quality of PoSh’s extracted scene graphs which offer complete coverage over all of the visual details specified in a generated description and its accompanying reference description.  The strength of these scene graphs is also reflected when evaluating PoSh against human judgments (Table 2 and Table 3), as error filled scene graphs would degrade its performance relative to its peers.
>
> We have updated our manuscript to include this additional human evaluation (lines 476 - 480, 1287 - 1293, 1303 - 1310).

---

> ### Author Response · Authors · 2025-11-25
> **Response to Reviewer 6TR7 (part 2/2)**
>
> > ### Expansion of detailed image description to artwork
>
> While we agree that images of natural scenes are more common, we expand detailed image description to visual art for several reasons.
>
> First, as enumerated in table 1 (lines 146-161), existing detailed image description benchmarks already focus on web imagery that skews natural.  For example, the images in CapArena are drawn from DOCCI where one of the authors used photographs that they had taken themselves, described by paid crowd workers.  In contrast, DOCENT features imagery curated and described by experts at the National Gallery of Art and published as part of their Open Data Initiative (lines 261-262), expanding detailed image description to a new image domain with extremely high quality data.
>
> Second, this expansion is meaningful – many museums lack high quality assistive text on their websites due to the cost of producing it manually [2].  DOCENT, whose references were written according to expert-informed assistive text guidelines (line 275), allows contributions from the broader research community to this socially impactful application.
>
> Finally, we want to highlight the special qualities of visual art that make it interesting for multimodal AI research – **art exhibits more complexity than natural images** and is therefore of particular interest for detailed image description.  **Compared to the natural images in CapArena, the references in DOCENT are more than twice as long and have scene graphs that are nearly three times as large (Table 1, line 160 vs 159); as a percentage, twice as many of its images contain two or more objects and 10 times as many of its images contain two or more people (line 372-373)**.   This complexity stresses the ability of statistical AI systems.  In contrast, some of the natural images in CapArena are quite trivial – for example, one image features only a plain clock.  Thus, detailed description of artwork is a useful testbed for tracking progress in VLMs.
>
> We want to emphasize that though DOCENT is focused on visual art, we evaluate PoSh on natural images too, validating its robustness.  It performs well on the relatively simple images in CapArena as well (Table 3, line 452; line 463).
>
> [2] https://www.artnews.com/art-in-america/columns/the-met-mca-chicago-blind-access-alt-text-park-mcarthur-shannon-finnegan-1202677577/
>
> > ### Replicability
>
> One of the core motivations behind the development of PoSh was the introduction of a replicable metric for evaluating detailed image descriptions, in contrast to the growing popularity of GPT5-as-a-Judge.   PoSh achieves this through the use of open models.  Like other open weight approaches that rely on probabilistic models (e.g., LLaVa Critic), these models may produce slightly different results on different hardware.  However, **using PoSh with the same GPU / version of CUDA / library versions (as specified in our repo) results in the same numbers**.
>
> Recently, vLLM (the framework we use to implement PoSh) added support for batch invariance [3].  This ensures replication on the same hardware with different inference hyperparameters (for example, with prefix caching disabled or with restricted GPU memory utilization).  To strengthen PoSh’s replicability further, we have added support for batch invariance to our implementation and we updated our manuscript to reflect this perfectly replicable evaluation setting (lines 541 - 542).
>
> [3] https://docs.vllm.ai/en/latest/features/batch_invariance
>
> Thank you for requesting a more complete explanation of what distinguishes PoSh from DSG, suggesting a more thorough evaluation of PoSh’s scene graphs, inquiring about the value of art as a testing domain in vision and asking for an explanation of what enables PoSh’s replicability.  We have updated our manuscript to answer all of your questions and believe our work is stronger for it.

---

### Author Response · Authors · 2025-12-03
**Summary for Area Chair**

We thank the reviewers for their thoughtful feedback which has substantially strengthened our submission.  Below, we summarize their reviews and the changes/responses we made to address their questions during the rebuttal period (these are also reflected in our manuscript where all updates are in blue).  For additional details, please consult our full responses to each individual reviewer.

Reviewer 6TR7 (score: 4, confidence: 4) [\[link\]](https://openreview.net/forum?id=UBhY1c4r2W&noteId=uP9ufmIwO8):
- *strengths*: found our paper to be “**well-written**”, our area of focus to be “**non-trivial**”, our proposed “**metric… [to be] important**”, our emphasis on metric grounding to be well-placed and our dataset DOCENT to be of “**quality [that is] well controlled**”
- *rebuttal responses*: we added a detailed comparison showing **PoSh outperforms the Davidsonian Scene Graph metric (which performs near chance)**, validated our extracted scene graphs against hand annotations (**F1 = 0.892**), clarified the value of an extension of detailed image description to art which is **much more visually complex than the images in existing benchmarks** (Table 1, line 161 vs line 154; lines 372 - 373), and improved the replicability of our results through **implementation of batch invariance**

Reviewer hCTY (score: 8, confidence: 4) [\[link\]](https://openreview.net/forum?id=UBhY1c4r2W&noteId=sZfXrgpfcy):
- *strengths*: found our paper had “**numerous strengths**” including making an “**important contribution**” to the “**important [task] of long-form image captioning**”, our emphasis on interpretability and granularity to be well-placed, our evaluation of PoSh to be “**very complete**” (including its value as a reward function) and our introduction of DOCENT to be a useful testbed for evaluating future metrics / the generation quality of future VLMs in a “**novel domain**”
- *rebuttal responses*: we validated our extracted scene graphs (**F1 = 0.892**) and our scene graph verification (**F1 = 0.852**) against hand annotations and clarified PoSh’s potential as a scalable reward function for post-training using synthetic detailed image descriptions

Reviewer cNnP (score: 6, confidence: 4) [\[link\]](https://openreview.net/forum?id=UBhY1c4r2W&noteId=jvCIPdZDRD):
- *strengths*: found the task we focus on to be “**important**”, our emphasis on interpretability to be valuable, our dataset to reflect “**massive manual effort**”, our “**open-source metric and dataset to be contributions that benefit the community**”, and our evaluation of our metric to be thorough
- *rebuttal responses*: we clarified naming for RL results, explained the value of an extension of detailed image description to art which is **much more visually complex than the images in existing benchmarks** (Table 1, line 161 vs line 154; lines 372 - 373), explained that PoSh is especially well suited for images that exhibit such complexity which are of particular interest in detailed image description (**it’s the top open-source metric on DOCENT and outperforms the larger LLaVaCritic-72B on visually complex images in CapArena**); however, their request for additional RL experiments was infeasible due to compute constraints

Reviewer XyHZ (score: 4, confidence: 2) [\[link\]](https://openreview.net/forum?id=UBhY1c4r2W&noteId=SQb53V358c):
- *strengths*: found that our work “**addresses a clear gap in evaluating detailed image descriptions**”, is “**accessible and deployable by researchers with limited resources**”, and that our dataset “**enables evaluation for complex visual domains**”
- *rebuttal responses*: we added runtime measurements (**PoSh evaluates descriptions in 2 seconds versus 25 seconds for DCScore or 18 minutes for humans**) and clarified that reference-based metrics like PoSh cannot be used for data curation; however, **their request to evaluate PoSh on SG-LLaVa could not be completed as this model remains unpublished**

Thank you to our area chair for their efforts in keeping ICLR afloat this year – we really appreciate it!

---

### Meta-Review · Area_Chair_fVqz · 2026-01-07

**Summary:**

The submission propose PoSh, a metric for image description. It use scene graphs to guide LLM for judging the omissions and mistakes made in the generated prompt compare with the reference prompt. The graph is used for generating structured Query-Answer problems. The submission also provide DOCENT, a dataset comprising of 1,750 artworks with fine and coarse annotations. The benchmarks exhibits better correlation with human judgments. And it can also be used as an RL reward function.

The reviewers mainly concern on the scene graph extractor, the choice of using artworks, the computational overhead and the effectiveness in RL.

**Reviewer Concerns:**

Addressed Concerns:

Reviewer 6TR7:
  1. Reliance on scene graph model introduces inaccuracies and errors: Evaluating results show that the extractor can work at high accuracy.
  2. Clarifying of uniqueness (comparison to DSG): PoSh is designed for evaluating image description and DSG is designed for evaluating text-2-image generation. PoSh can process longer prompts, and works in a reference way.
  3. The choice of artworks: These artworks are of high quality and prompts are written according to expert-informed assistive text guidelines. Art exhibits more complexity than natural images and is special to community users.
  4. Repeatability: This is an implementation details related with architecture.

Reviewer hCTY:
  1. Reliance on scene graph model introduces inaccuracies and errors: Evaluating results show that the extractor can work at high accuracy.
  2. Too expensive to be used as RL reward: PoSh as a reward is expensive, but high quality synthetic detailed image descriptions could be possible in future.
  3. Performance of LLM-as-a-Judge’s QA answers: Verification component achieves strong alignment with labels.

Reviewer cNnP:
  1. Writing problem: addressed in revision.
  2. DOCENT vs CapArena: Artworks in DOCENT is more complexity than natural images in CapArena.

Reviewer XyHZ:
  1. Computational overhead: The tested evaluation efficiency is competitive compared with methods relying on GPT4.
  2. Lack of validation as a data curation tool for MLLM training: PoSh cannot be used for curating data since it is reference-based.

Remaining Concerns:

Reviewer cNnP:
  1. PoSh as a reward function: impossible to get result on additional models before the rebuttal period.
  2. Other caption benchmarks for validation: No other released benchmarks.

Reviewer XyHZ:
  1. Evaluation on other captioning models with structural priors:  SG-LLaVa is not accessible, and this is added as a limitation.

**Reviewer Scores:**

Reviewer 6TR7: rank 4, inactive in rebuttal period, would raise the rank for concerns get addressed (-> rank 6).

Reviewer hCTY: rank 8, inactive in rebuttal period, would remain the rank.

Reviewer cNnP: rank 6, inactive in rebuttal period, would remain the rank.

Reviewer XyHZ: rank 4, inactive in rebuttal period, would remain the rank for hanging concerns.

The benchmark proposed in the paper is reasonable in intuition, effectiveness and applicability, which would help to evaluate the image descriptor in more detailed and from multiple perspectives.

---

### Decision · Program_Chairs · 2026-01-26

Accept (Poster)